# α7 nicotinic acetylcholine receptors regulate radial glia fate in the developing human cortex

Tanzila Mukhtar [1,2,5] ✉, Clara-Vita Siebert[1,2,3,5], Yuejun Wang[1,2,5], Mark-Phillip Pebworth[1,2,4,5], Matthew L. White [1,2], Tianzhi Wu[1,2], Tan Ieng Huang[1,2], Guolong Zuo [1,2], Jayden Ross[1,2], Jennifer Baltazar [1,2], Varun Upadhyay[1,2], Merut Shankar[1,2], Li Zhou[1,2], Isabel Lombardi-Coronel[1,2], Ishaan Mandala[1,2], Manal A. Adam[1,2], Shaohui Wang[1,2], Qiuli Bi[1,2], Marco F. M. Hoekman[3], Jingjing Li [1,2] & Arnold R. Kriegstein [1,2] ✉

Prenatal nicotine exposure impairs fetal cortical grey matter volume, but the precise cellular mechanisms remain poorly understood. This study elucidates the role of nicotinic acetylcholine receptors (nAChRs) in progenitor cells and radial glia (RG) during human cortical development. We identify two nAChR subunits—CHRNA7 and the human-specific CHRFAM7A—expressed in SOX2+ progenitors and neurons, with CHRFAM7A particularly enriched along RG endfeet. nAChR activation in organotypic slices and dissociated cultures increases RG proliferation while decreasing neuronal differentiation, whereas nAChR knockdown reduces RG and increases neurons. Single-cell RNA sequencing reveals that nicotine exposure downregulates key genes in excitatory neurons (ENs), with CHRNA7 or CHRFAM7A selectively modulating these changes, suggesting an evolutionary divergence in regulatory pathways. Furthermore, we identify YAP1 as a critical downstream effector of nAChR signaling, and inhibiting YAP1 reverses nicotine-induced phenotypic alterations in oRG cells, highlighting its role in nicotine-induced neurodevelopmental pathophysiology.

Neurodevelopmental and neuropsychiatric diseases including autism, schizophrenia, and attention deficit hyperactivity disorder (ADHD) have been linked to maternal smoking during pregnancy[1–3]. Despite the risks to fetal development, a global average of 1.7–21% of pregnant mothers continue to smoke[4,5]. Not only are the by-products of cigarette smoke harmful, but the active compound nicotine poses significant risks, raising concern about the safety of nicotine replacement therapy during pregnancy[6,7]. Nicotine, an exogenous ligand of nicotinic acetylcholine receptors (nAChRs), competes with the

endogenous neurotransmitter acetylcholine (ACh). Notably, nicotine exhibits a higher binding affinity for nAChRs than ACh, potentially amplifying its impact on the developing brain[8]. Evidence suggests that prenatal smoking or smokeless nicotine exposure leads to reduced cortical gray matter volume, cortical thinning, and behavioral abnormalities in offspring[6,7,9–12]. The Generation R study that followed children of 113 mothers who smoked during pregnancy found behavioral abnormalities and changes in brain morphology by school-age[12]. Despite consistent pathological effects, the molecular mechanism by

[1]Department of Neurology, University of California, San Francisco, CA, USA. [2]The Eli and Edythe Broad Center of Regeneration Medicine and Stem Cell Research, UCSF, San Francisco, CA, USA. [3]Swammerdam Institute for Life Sciences, University of Amsterdam, Amsterdam, The Netherlands. [4]Allen Institute of Immunology, Seattle, WA, USA. [5]These authors contributed equally: Tanzila Mukhtar, Clara-Vita Siebert, Yuejun Wang, Mark-Phillip Pebworth. ✉e-mail: Tanzila.mukhtar@ucsf.edu; Arnold.Kriegstein@ucsf.edu

which smoking causes cortical and behavioral abnormalities remains elusive.

Nicotine exposure has a variety of effects on brain development in rodents and non-human primates that manifest in particular brain regions (including neocortex) and at specific developmental time points[13–16]. NAChRs are expressed in mice as early as embryonic day 10 (E10) when progenitor cells are the most prevalent cells in the cortex[17]. Moreover, most E10-E11 neurons show functional responses to the broad AChR agonist, ACh, as well as the more specific agonist, nicotine[17]. Rodent studies have shown that nicotine can affect neurogenesis and differentiation as well as synaptic development[18–21]. In vivo studies in rodents show an increase in adult SVZ neurogenesis upon nicotine exposure mediated by FGF2 signaling[22,23]. However, non-human primates, such as Rhesus monkeys, provide a closer approximation of human brain development than rodent models. Nicotine administered to pregnant Rhesus monkeys at plasma levels comparable to human smokers alters the expression of α4β2 and α7 nAChRs and impairs neurite formation in the cortex of their affected offspring[15]. Despite the diverse effects nicotine can have on animal models, consequences of nicotine exposure on early developing human cortex remain unclear.

NAChR receptors are pentameric ligand-gated ion channels formed from a combination of subunits including α2-7, 9, 10 that play crucial roles in neural maturation and synaptic transmission[13,24]. The α7 nAChR is formed as a homomeric pentamer of the α7 subunit that is encoded by the CHRNA7 gene. This receptor plays a crucial role in cognitive function and has been implicated in schizophrenia[13,25,26]. During evolution, a partial duplication of the CHRNA7 gene has led to the formation of the human-specific gene CHRFAM7A[24]. CHRFAM7A encodes for the peptide subunit dupa7 that assembles with the native α7 subunit[27,28]. Microdeletions in CHRNA7 and CHRFAM7A have been associated with schizophrenia and bipolar disorder[24,29,30]. Despite the importance of nAChRs in human cognition and disease[27,28,31,32], the expression and roles of CHRNA7 and CHRFAM7A during human cortical development are not well understood.

Here we investigate the role of cholinergic signaling mediated through ACh or nicotine, and the role of CHRNA7 and CHRFAM7A in human radial glia (RG) and progenitor cells during cortical development. We use a range of model systems, including ex vivo organotypic slice cultures from primary developing human cortical samples, dissociated primary cortical cell cultures, and induced pluripotent stem cell (iPSC)-derived cerebral organoids.

## Results

### Cholinergic fibers and outer radial glia (oRG) colocalize in the developing human cortex

Cholinergic signaling in the developing cerebral cortex is largely mediated through cholinergic fibers from the nucleus basalis of Meynert, a cluster of cholinergic neurons in the basal forebrain[33], that innervate the cerebral cortex and express choline acetyltransferase (ChAT) and acetylcholinesterase (AChE)[34,35]. In humans, cholinergic fibers project towards the cerebral cortex as early as gestational week 12 (GW12)[34]. We sought to characterize the cortical location of cholinergic fibers relative to the RG-rich germinal zone (GZ). RG residing along the lateral ventricles are the principal stem cells of the developing human cortex, giving rise to oRG cells, intermediate progenitor cells (IPCs), neurons, and glia[36]. Humans have large numbers of oRG cells compared to mice, and they are thought to serve as key drivers of the expansion of the neocortex, particularly contributing to upper layer neurons[36,37]. We performed co-immunostaining for AChE, HOPX (a marker of oRG cells), and Vimentin (VIM, a marker of RG and oRG fibers), in coronal sections of primary developing human prefrontal cortex at GW15, and GW22 (Fig. 1a). AChE was enriched in the outer subventricular zone (oSVZ), cortical plate (CP), and marginal zone

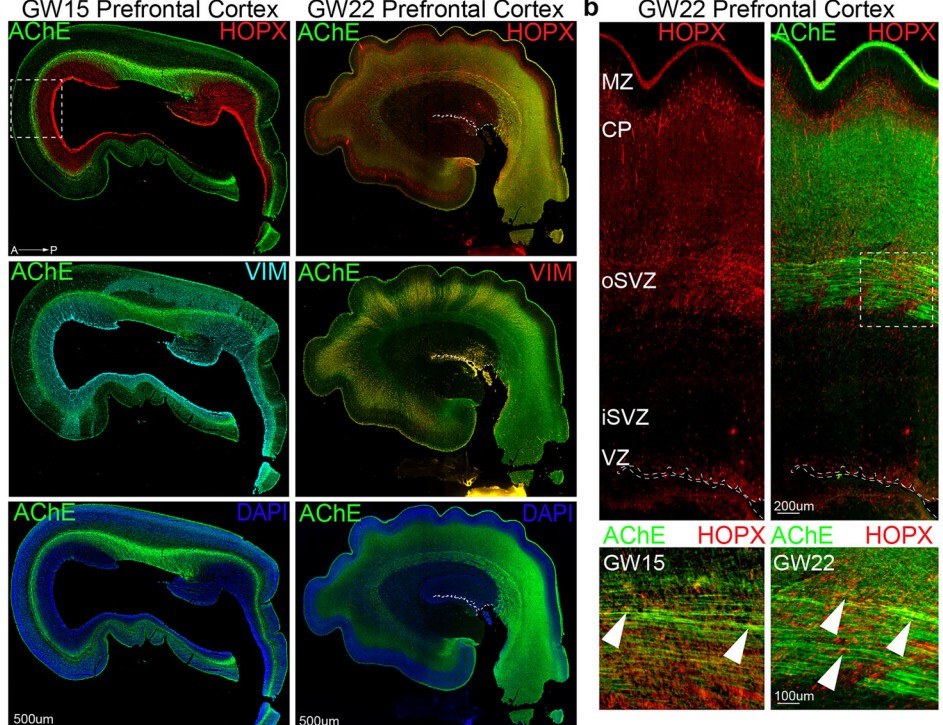

**Fig. 1 | Acetylcholinesterase (AChE) is expressed in the developing prefrontal cortex throughout cortical development (GW15-22). a** Representative images showing immunostaining of AChE+ cholinergic fibers, HOPX+ outer radial glia and vimentin (VIM)+ radial glia fibers at GW15, and GW22. Scale bar = 500 μm, *n* = 3. **b** AChE co-localizes with HOPX+ outer radial glia in the oSVZ. Representative images of GW15 and GW22 prefrontal cortex at higher magnification show overlap of HOPX+ oRG and AChE+ cholinergic fibers. Arrowheads show co-expression of HOPX and AChE. Scale bar = 200 μm and 100 μm, *n* = 3. Abbreviations: A-P (anterior-posterior), ventricular zone (VZ), inner subventricular zone (iSVZ), outer subventricular zone (oSVZ) and Cortical plate (CP).

(MZ), and colocalized with HOPX+ oRG cells and VIM+ radial fibers (Fig. 1b). We also observed similar patterns of AChE expression in GW15 and GW22 motor cortex and visual cortex (Supplementary Fig. 1a, b). These similar patterns across different cortical areas suggest a pervasive role of cholinergic signaling during cortical development.

## CHRNA7 and CHRFAM7A receptor subunits are expressed in major cortical progenitor niches

We hypothesized that due to the co-localization of AChE+ fibers and oRG cells, that ACh might functionally interact with oRG cells in the oSVZ progenitor niche. To investigate the cell types potentially responsive to cholinergic signals, we used a single-cell RNA-seq (scRNAseq) dataset of developing human cortex (hgwdev-max.soe.ucsc.edu/tsneViewer/dev3/) and verified the RNA expression of four nAChRs subunits: CHRNA7; CHRFAM7A; CHRNB1; and CHRNB2. All four subunits, though mostly studied in neurons, were also expressed by RG, dividing progenitors, and oRG cells, as well as early neurons, with CHRNA7 and CHRFAM7A showing more abundant expression in progenitors than CHRNB1 and CHRNB2 (Supplementary Fig. 2a). To validate translation of RNA into functional receptors, we performed Western blotting on protein lysates isolated from either whole cortices or the GZ of primary developing tissue obtained by microdissection. We collected two or three biological replicates per preparation type, spanning periods of high neurogenesis (GW15-GW19) and gliogenesis (GW20-GW23). We observed protein expression of CHRNA7, CHRFAM7A, CHRNB1 and CHRNB2 across all time points in whole cortex lysates and enriched GZ samples (Supplementary Fig. 2c–e).

To further validate cell-type-specific receptor expression in primary cortical sections (GW17-23), we performed RNAscope (for CHRNA7) and BaseScope (for CHRFAM7A) and co-immunostained for the progenitor cell marker, SOX2 (Fig. 2a)[38]. We observed high expression of CHRNA7 in the ventricular zone (VZ) and CP, and colocalization with both SOX2+ and SOX2- cells indicating broad expression in different cell types (Fig. 2a). CHRFAM7A RNA was highly expressed in the apical endfeet of RG and in the CP at GW17 (Fig. 2a″). Both subunits were expressed in SOX2+ oRG cells (Fig. 2b, c)[37]. Immunostaining for CHRNA7 and CHRFAM7A also validated protein expression of these subunits in SOX2+ progenitors within the VZ and in NEUN-expressing neurons in the CP, with lower expression in the oSVZ (Fig. 2b, c and Supplementary Fig. 3). Expression was also observed in ZO-1 + RG apical endfeet contacting the ventricle (Supplementary Fig. 2b), an expression pattern that might indicate a role for cholinergic signaling mediated by CSF. Choline is a selective activator of α7 nAChRs, that have lower affinities than other nAChRs for nicotine and ACh[13,39]. Choline is present in high concentrations in the developing fetus and is a source of cholinergic signaling during human cortical development[40,41]. These results indicate that RG and oRG cells express a variety of specific nAChRs.

## IPSC-derived cortical organoids display nAChR expression patterns observed in primary tissue

To further study the observed expression patterns of CHRNA7 and CHRFAM7A in neuronal lineage cells, we utilized iPSC-derived cortical organoids (Table 1). Cortical organoids are 3D spheres derived from pluripotent stem cells through the addition of morphogens and growth factors that mimic in vivo developmental signals[42,43]. We found CHRNA7 and CHRFAM7A protein expressed in organoids across a developmental time-course corresponding to progenitor expansion (week four), early neurogenesis (week seven), peak neurogenesis (week nine) and early gliogenesis (week 13) (Supplementary Fig. 2f)[44]. Comparable to primary tissue, we observed high CHRFAM7A expression in KI67+ cells in the progenitor-rich neural rosettes, including along the apical border, at weeks 7–9 (Fig. 2d, d′). By weeks 13–15 we observed the expression of CHRFAM7A in SOX2+ and NEUN+ cells

(Fig. 2e). Thus, cortical organoids have similar expression patterns for CHRNA7 and CHRFAM7A as primary tissue and are a reasonable model system to investigate cholinergic signaling.

## Activation of nAChRs promotes the proliferation of progenitors while reducing the number of neurons in the developing human cortex

To investigate the functional implications of cholinergic signaling, we treated organotypic cortical slice cultures at GW18-19 ($n = 3$) with two broad nAChR agonists; acetylcholine and nicotine, and included an EdU pulse 24 h before fixation (Fig. 3a). Upon ACh or nicotine treatment, the number of SOX2+ progenitors increased significantly compared to untreated (UT) control (Fig. 3b, c). Likewise, KI67+ proliferating cells increased in ACh and nicotine conditions (Fig. 3b, c), consistent with an observed increase in EdU-positive cells (Fig. 3b, c). EOMES+ intermediate progenitor cells (IPCs) increased post nicotine treatment, while remaining unchanged after ACh, suggesting different responses to ACh and nicotine (Fig. 3d, e). Nicotine primarily targets nAChRs, whereas ACh interacts with both nicotinic and muscarinic acetylcholine receptors[45]. Additionally, we observed a significant increase in HOPX+ oRG cells in both conditions compared to controls (Supplementary Fig. 4a, b). The balance between progenitor self-renewal and neuronal differentiation is tightly regulated in the developing cortex[36]. During the second trimester (GW15-19), RG and oRG cells generate large numbers of cortical neurons[36]. We investigated if the increased self-renewal of RG, oRG, and IPCs impacts neuronal differentiation. We found an increase in double-positive KI67 and EdU cells following ACh and nicotine treatment, indicating extended cell cycle length (Supplementary Fig. 4a, b). Indeed, the increase in progenitor cells was associated with a reduction in NEUN+ neurons affecting both deep (CTIP2 + ) and upper (SATB2 + ) layer neurons (Fig. 3d, Supplementary Fig. 4a, a′, b). The NEUN+ neurons were reduced with ACh and nicotine, impacting deep layer CTIP2+ neurons in the IZ. To investigate the impact of ACh and nicotine on gliogenesis, we quantified astrocytes (labeled with S100B) and oligodendrocytes (labeled with OLIG2) (Supplementary Fig. 4c, d). We found no significant changes in S100B and OLIG2 expression during the 5-day chase after agonist treatment. However, previous studies have observed changes in astrocytes and oligodendrocytes following nicotine exposure, possibly attributed to different exposure paradigms used[46,47]. We also observed no significant change in DLX5 interneurons or cell death across conditions (Supplementary Fig. 4e, f). While we assessed DLX5-positive cells in the cortex, we did not investigate the effects of nicotine exposure on the ganglionic eminence, which is the primary source of cortical interneurons.

To validate the role of the α7 nAChR in these phenotypic changes, we treated organotypic slices with a highly specific agonist, PNU, and included DMSO as a negative control. We observed that PNU was sufficient to increase SOX2 + , HOPX + , and KI67+ progenitors (Fig. 3e, f, Supplementary Fig. 4g, h). PNU increased SOX2+ cells, but did not reach statistical significance ($p = 0.054$). HOPX+ and KI67+ cells increased post PNU treatment. PNU did not significantly impact EOMES+ IPCs or EdU incorporation (Fig. 3e, f and Supplementary Fig. 4g, h). This suggests the α7 nAChR is involved in neurogenesis through activation of HOPX+ oRG cells. We observed a reduction in NEUN+ neurons with PNU, impacting CTIP2+ deep layer neurons, with no change in upper layer SATB2+ neurons (Fig. 3e and Supplementary Fig. 4h, i). DLX5 interneurons did not change in number and there was no increase in cell death (Supplementary Fig. 4j, k). Overall, these results show that broad activation of nAChRs through ACh or nicotine promotes progenitor cell fate at the cost of neurogenesis. Activation of the α7 nAChR recapitulates several of the phenotypic features observed with ACh or nicotine exposure, confirming a role of the α7 nAChR in progenitor maintenance, proliferation, and regulation of neurogenesis in developing human cortex.

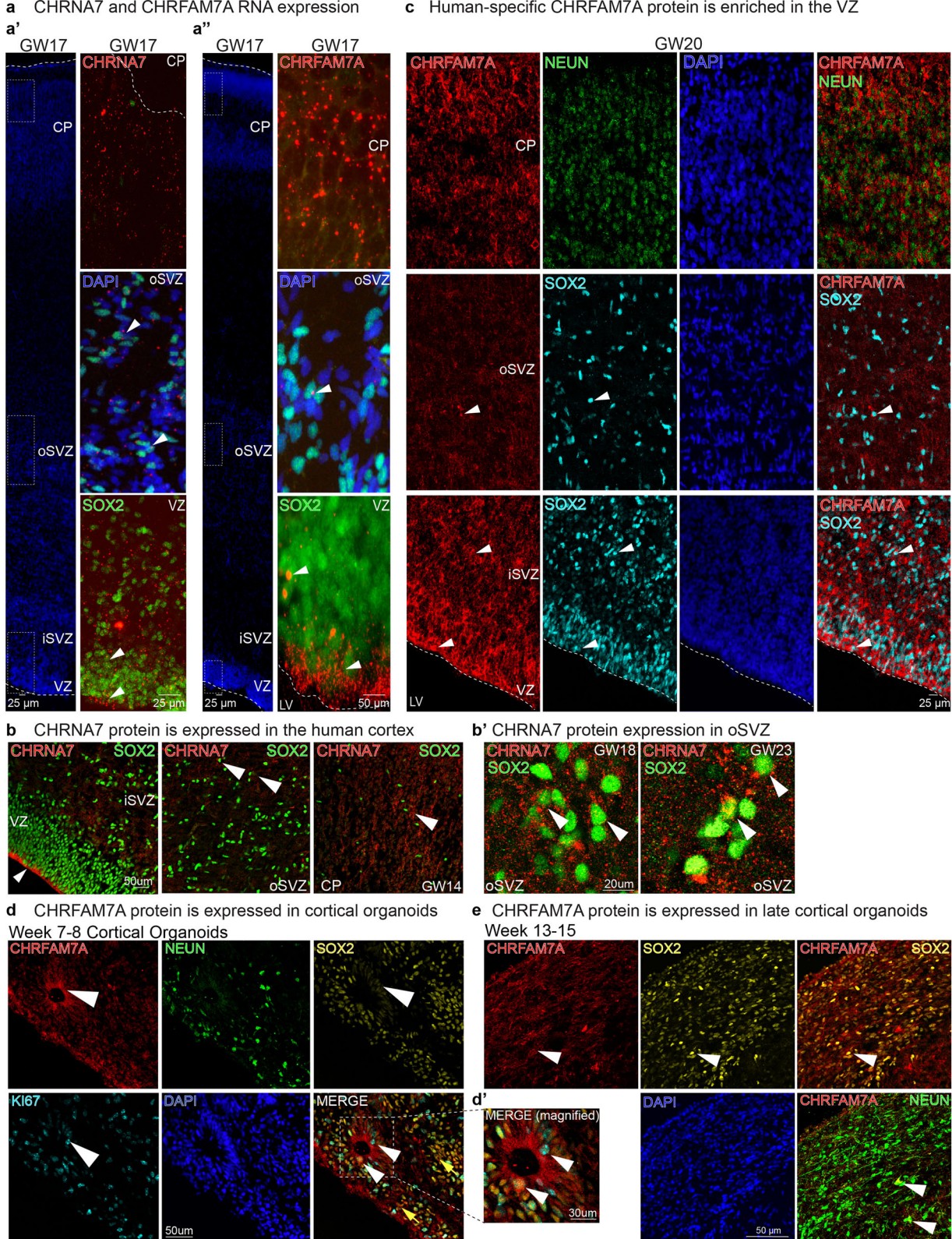

**Fig. 2 | Expression of Cholinergic receptor subunits CHRNA7 and CHRFAM7A in the developing cortex and cortical organoids. a, a', a"** CHRNA7 and CHRFM7A RNA expression. **a** RNAscope for CHRNA7, and **a'** BaseScope for CHRFAM7A, with coimmunostaining of radial glia marker SOX2 shows expression of both subunits in progenitor cells at GW17 (white arrowheads). CHRFAM7A RNA is enriched along the apical endfeet. **b, b'** Immunostaining of CHRNA7 and SOX2, in VZ, oSVZ and CP, shown in GW14, GW18, and GW23 cortical samples **c.** Human-specific CHRFAM7A protein is enriched in the VZ. Immunostaining of CHRFAM7A validates enrichment of CHRFAM7A in the apical endfeet of the developing cortex. Colocalization of CHRFAM7A protein with SOX2 is observed in the VZ and oSVZ (white arrows). **d, d', e** CHRFAM7A protein is expressed in ventricular RG in cerebral organoids. Immunostaining of CHRFAM7A in cortical organoids at weeks 7–9 and weeks 13–15 replicate primary tissue with expression of CHRFAM7A in SOX2+ progenitors and enrichment within neural rosettes. Abbreviations: ventricular zone (VZ), inner subventricular zone (iSVZ), outer subventricular zone (oSVZ) and Cortical plate (CP), n = 3.

**Table 1 | Pluripotent stem cell lines**

| Cell line | Designation | Source | Identifiers | Additional Information |
|---|---|---|---|---|
| Cell line (*Homo sapiens*) | H1/WA01 Embryonic stem cell line | WiCell | RRID:CVCL_9771 | Male |
| Cell line (*Homo sapiens*) | WTC-11 induced pluripotent stem cell line | Conklin Laboratory (Gladstone Institutes) | GM25256 | Male |
| Cell line (*Homo sapiens*) | 1323-4 induced pluripotent stem cell line | Conklin lab (Gladstone Institute) | RRID:CVCL_0G84 | Female |

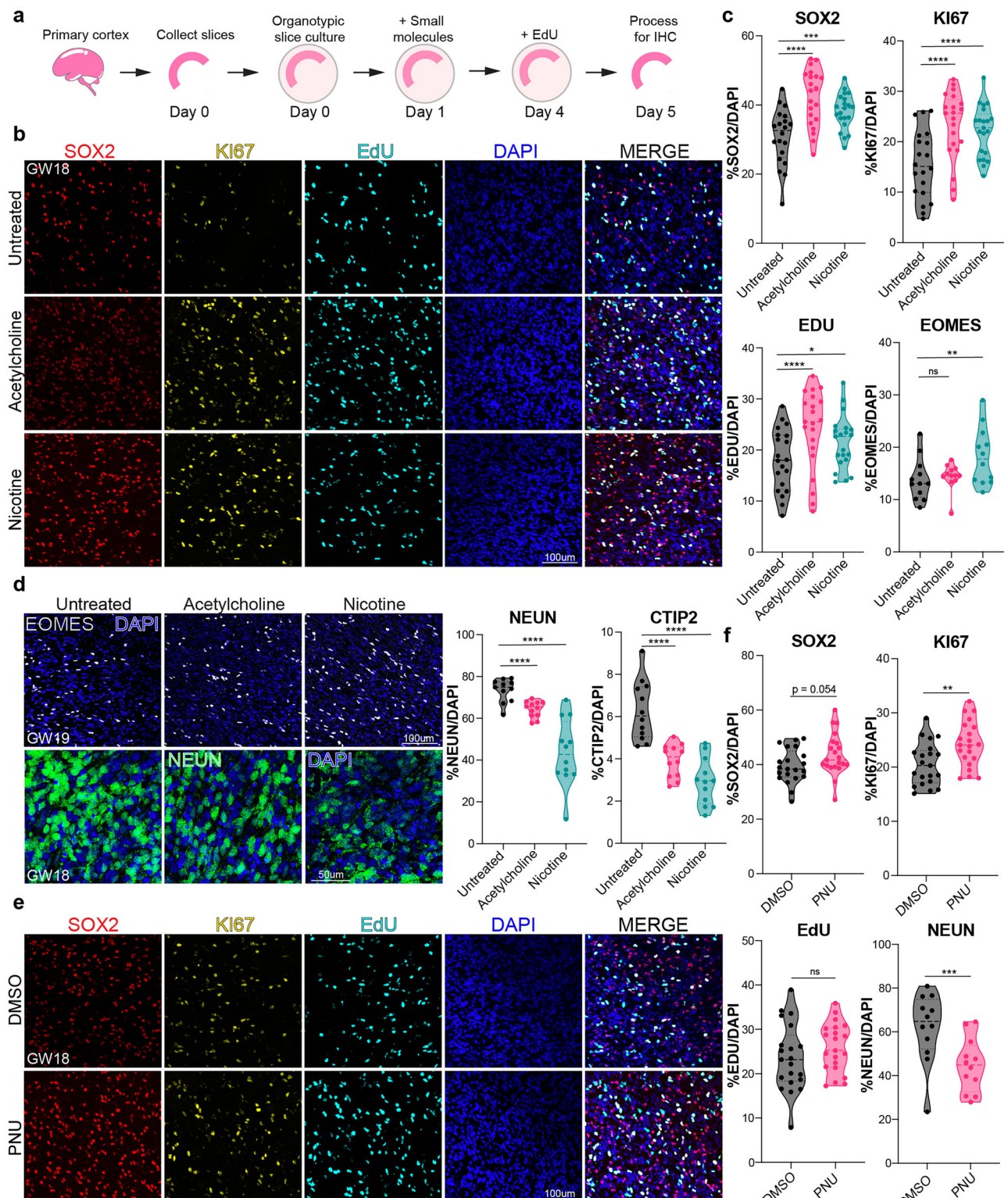

**Fig. 3 | Activation of cholinergic signaling in primary cortical organotypic slices by agonist treatment increases progenitor cells and inhibits neuronal differentiation. a** Scheme showing treatment of cortical organotypic slices with broad (ACh) and specific (nicotine, PNU) agonists for 5 consecutive days. EdU was added on Day 4 and slices fixed with 4% PFA on Day 5 for Immunohistochemistry (IHC). **b** Representative images showing GW18 organotypic slices post ACh and nicotine treatment, immunostained for SOX2 +, KI67 +, EdU +, and co-stained with DAPI. Scale bar = 100μm. **c** Quantification of SOX2, KI67, EdU, and EOMES over total DAPI, post broad agonist treatment, $n = 3$. Upon ACh or nicotine treatment, the number of SOX2+ progenitors increased significantly to 42.1% ($p = 1.12E\text{-}06$), and 38.1% ($p = 0.0007$) of DAPI+ cells respectively, compared to 30.8% in untreated (UT) control. KI67+ proliferating cells increased to 23.3% (ACh, $p = 1.61E\text{-}06$) and 22.1% (nicotine, $p = 2.66E.05$), compared to 15.5% in controls. We observed an increase in EdU+ cells (24% for ACh, $p = 8.61E\text{-}05$; 21.4% for nicotine, $p = 0.016$; 18% UT control). **d** Representative immunostaining for EOMES and NEUN in organotypic slices

treated with agonists. Scale bar = 100 μm. EOMES+ intermediate progenitor cells (IPCs) increased post nicotine to 18.2% ($p = 5.47E\text{-}05$) from 13.2%, while remaining unchanged after ACh. Quantification for NEUN and CTIP2 shows changes in neuronal proportions post agonist treatment. NEUN+ neurons were reduced to 64.3% with ACh ($p = 0.02$) and reduced to 43.5% ($p = 2.93E\text{-}09$) with nicotine, compared to 73.4% in the UT control. Deep layer CTIP2+ neurons in the IZ were reduced from 6.3% to 3.9% ($p = 1.25E\text{-}05$) with ACh, and 2.9% ($p = 2.14E\text{-}08$) with nicotine. **e** Immunostaining of SOX2, KI67, and EdU post PNU treatment. Scale bar = 100 μm. **f** Quantification for markers post specific activation of α7 nAChR. PNU increased SOX2+ cells to 44% compared to 39% in the DMSO control, but did not reach statistical significance ($p = 0.054$). KI67+ cells increased to 24.2% ($p = 0.003$) from 20.5%. PNU did not significantly impact EdU incorporation. NEUN+ neurons reduced from 63.6% in DMSO control to 45.5% ($p = 0.001$) with PNU, ($n = 3$), Source Data, two-sided t-test, $*p = 0.05$, $**p = 0.01$, $***p = 0.001$, $****p = 0.0001$. ns not significant.

## Activation of nAChRs in cortical organoids partially phenocopies activation in primary tissue

Cortical organoids provide a useful model system to manipulate signaling effectors. To replicate our observations from primary tissue, we treated cortical organoids derived from three iPSC lines (H1, 1323-4 and WTC-11) with ACh, nicotine or PNU after seven weeks of differentiation, a time-point corresponding to a period of neurogenesis[44] (Fig. 4a). We observed a significant increase in SOX2+ progenitors with both ACh and nicotine and a simultaneous reduction in NEUN+ neurons (Fig. 4b–d). Specifically, SATB2+ upper layer neurons decreased in both conditions, while the number of CTIP2+ deep layer neurons was reduced in nicotine treated organoid cultures only (Fig. 4b–d). We observed a trend toward an increase of SOX2+ progenitors and a reduction in NEUN+ cells in PNU treated organoids (Fig. 4e, f).

The generation of deep-layer CTIP2+ neurons in organoids starts at stages prior to seven weeks, and the absence of changes in CTIP2+ neurons in ACh treated organoids might be due to the developmental window when the organoids were treated[42]. To test the impact of AChR agonists on CTIP2+ deep-layer neurons, we treated cortical organoids derived from three different cell lines at a younger age (week 5) for four days (Supplementary Fig. 5a). We observed a consistent increase in SOX2+ progenitors with ACh and nicotine (Supplementary Fig. 5a–c) accompanied by a simultaneous reduction in NEUN+ and CTIP2+ cells (Supplementary Fig. 5a–c).

We attempted to rescue the phenotypes by adding the α7 nAChR antagonist, α-Bungarotoxin (BTX)[48]. There were no phenotypic changes in progenitors or neurons with BTX, but in combination with the agonists (ACh or nicotine), BTX was able to normalize the percentages of SOX2 +, NEUN+ and CTIP2+ cells (Supplementary Fig. 5b, c). We observed significant changes in NEUN+ cells, compared to UT controls. CTIP2+ cells changed slightly (Supplementary Fig. 5b, c). This rescue suggests that blocking the α7 nAChR receptor can at least partially reverse the observed agonist-mediated phenotypic changes, and activation of β-receptors may partially explain the incomplete rescue. Thus, the observed effects are likely mediated, in part, by the α7 nAChR. We also tested PNU in early cortical organoids but did not observe a significant phenotype in progenitors or neurons, suggesting potential mechanistic differences between early and late neurogenic organoids downstream of agonist treatment (Supplementary Fig. 5d, e). We noticed no changes in oRG cells in the presence of agonist (Supplementary Fig. 5f), although there are very few HOPX+ oRG cells at this stage[44]. These observations highlight that broad activation of nAChRs in early organoids recapitulates several phenotypic changes in fetal progenitors, and can impact CTIP2+ early born deep-layer neurons. Temporal patterning and cell fate decisions are critical during cortical development and nAChRs appear to regulate these fundamental processes.

## Loss-of-function of CHRNA7 and CHRFAM7A show inverse phenotypic changes compared to receptor activation

We conducted loss-of-function experiments to further test the role of nAChRs during cortical development. Lentiviral vectors expressing shRNAs and RFP were used to infect dissociated primary cortical cultures derived from GW14–18 tissue samples ($n = 4$) (Supplementary Fig. 6a). Using RT-qPCR, we observed a 43% and 54% reduction in mRNA expression of the respective subunits in shCHRNA7 and shCHRFAM7A treated cultures compared to scrambled controls (shCONTROL) (Supplementary Fig. 6b). We also tested the lentiviral infection efficiency in organotypic slices and observed a comparable number of RFP+ cells across conditions (Supplementary Fig. 6b).

We treated organotypic slice cultures (GW15, 17 and 21, $n = 3$) with shRNAs for seven days and used immunostaining to analyze potential phenotypic changes (Fig. 5a). We observed a reduction of SOX2+ progenitors upon loss-of-function of CHRNA7 and CHRFAM7A, an effect opposite to agonist treatment (Fig. 3b, c). We also observed a reduction in KI67+ dividing progenitors (Fig. 5b–d). We found no change in EOMES+ IPCs (Fig. 5c, d). The number of NEUROD1+ newborn neurons, increased upon knockdown of CHRFAM7A while NEUN+ neurons increased upon knockdown of both CHRNA7 and CHRFAM7A (Fig. 5c, e). To investigate the role of nAChRs in gliogenesis, we performed immunostaining for S100B and OLIG2, markers of astrocytes and oligodendrocytes, respectively. We found a reduction in astrocytes and a significant decrease in oligodendrocytes (Fig. 5c, f). Our findings suggest that both CHRNA7 and CHRFAM7A subunits are required for RG maintenance, proliferation and gliogenesis. The human specific CHRFAM7A appears to play a critical role in neurogenesis, consistent with its high expression in ventricular RG.

## RNA sequencing reveals the molecular changes associated with nAChR signaling

To derive a mechanistic understanding about cholinergic signaling in progenitor cells, we performed bulk RNA-sequencing following nAChR knockdown. We microdissected the GZ from GW15-23 cortical slice cultures ($n = 4$, biological replicates). Since choline is present at high concentrations in the developing fetal brain[13,39,41], we hypothesized that treatment with shRNAs against CHRNA7 or CHRFAM7A would allow us to explore loss-of-function of nAChRs (Supplementary Fig. 6a, b). Knockdown of CHRNA7 upregulated CACNA1A, a gene involved in neurogenesis and implicated in autism spectrum disorder (ASD) and epileptic encephalopathy[49] (Supplementary Fig. 6c). We also found upregulation of UNC5C, known to be involved in axon extension and migration[50,51]. CHRNA7 loss upregulated GABRG3, a gene mediating neuronal inhibition that has been associated with ASD[52]. CHRNA7 knockdown caused a downregulation of GNB1, which is a heterotrimeric G-protein that integrates signals between receptors and effector proteins. GNB1 is associated with Global Developmental

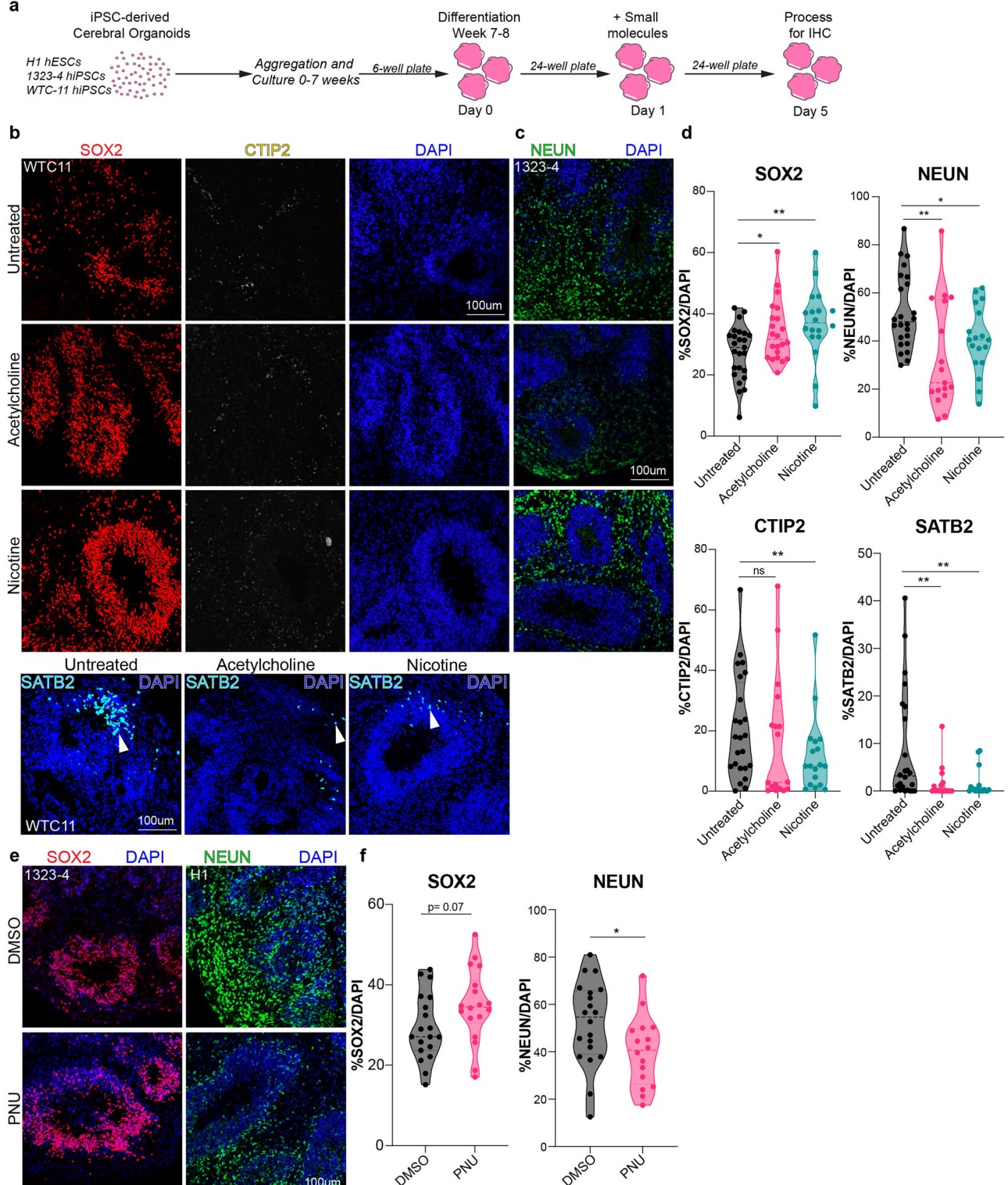

**Fig. 4 | Activation of cholinergic signaling in cortical organoids shows comparable trends to primary tissue. a** Scheme of cortical organoid differentiation using the Modified Sasai Protocol[44], followed by treatment with broad (ACh) and specific (nicotine, PNU) agonists. Three control cortical organoid lines were treated at differentiation Week 7–8, for 5 consecutive days, PFA fixed on Day 5 and processed for IHC. **b** Representative images of cortical organoids, immunostained for SOX2, CTIP2 and SATB2 co-stained with DAPI. Scale bar = 100 μm. **c** Representative images of cortical organoids immunostained for NEUN and co-stained with DAPI. Scale bar = 100 μm. **d** Quantification for SOX2, NEUN, CTIP2 and SATB2. SOX2+ cells in ACh and nicotine increased to 37% (*p* = 0.01) and 34% (*p* = 0.001), respectively, compared to 27.4% in UT controls. The number of NEUN+ neurons was reduced

from 52% to 34% (*p* = 0.002) in ACh and 40.4% (*p* = 0.02) in nicotine. SATB2+ neurons changed from 8.5% in control to 1.5% (*p* = 0.007) and 1.4% (*p* = 0.007) with ACh and nicotine treatment, respectively. **e** Representative images of cortical organoids post treatment with specific agonist, PNU and DMSO control stained for SOX2, NEUN, and DAPI. Scale bar = 100 μm. **f** Quantifications for SOX2 and NEUN post PNU treatment. In PNU treated organoids, SOX2+ progenitors increased from 29% in DMSO to 35% (*p* = 0.07) in PNU-treated conditions. NEUN+ cells were reduced to 40% (*p* = 0.02) in PNU treated cortical organoids, compared to 52% in DMSO controls, *n* = 3, Source Data, two-sided t-test, *\*p* = 0.05, *\*\*p* = 0.01, *\*\*\*p* = 0.001, *\*\*\*\*p* = 0.0001. ns not significant.

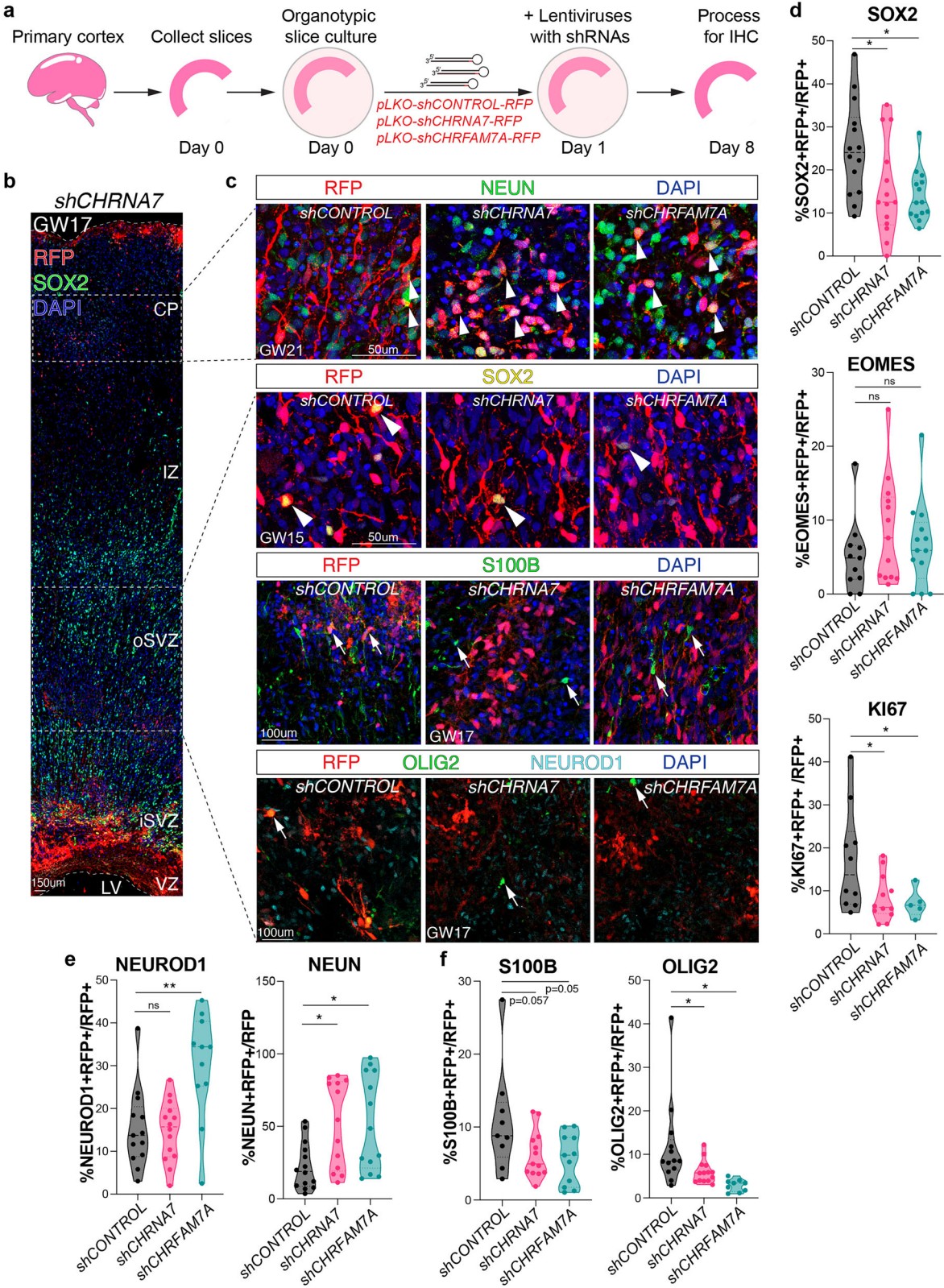

Delay, a disorder characterized by intellectual disability and seizures[53]. Knockdown of CHRNA7 also caused a reduction in PSAT1, a gene whose expression is reduced in schizophrenia[54]. To identify which biological processes are most affected by knockdown of CHRNA7, we conducted GO analysis and found basic processes such as semaphorin receptor signaling, chemokine receptor binding, and STAT pathway among upregulated processes. We also observed downregulation

of glycerolipid metabolism, phospholipid metabolism, and acetylcholine-gated cation channel activity, consistent with CHRNA7 regulating a variety of basic cellular processes in cortical progenitors (Supplementary Fig. 6d).

We observed downregulation of multiple genes upon knockdown of CHRFAM7A, including PPP1CA, IL6ST, CALR, and RCN2 (Supplementary Fig. 6c). IL6ST, highly enriched in neural progenitors, is a

**Fig. 5 | Loss-of-function of CHRNA7 and CHRFAM7A receptor subunits shows opposite phenotypic changes compared to cholinergic activation.**
**a** Experimental design for organotypic slice culture preparation and treatment with lentiviral mediated-shRNAs-directed against CHRNA7 or CHRFAM7A, and a scrambled control. Samples were fixed after 7-days with 4% PFA and processed for IHC. **b** Tile scan of example shCHRNA7 sample highlighting the regions used for imaging and quantifications **c**. Representative images of lentiviral infected, RFP+ cells, co-immunostained with NEUN+ (arrows) from GW21 with shCONTROL, shCHRNA7 and shCHRFAM7A. Scale bar = 50 μm. Representative images of lentiviral infected, RFP+ shCONTROL, shCHRNA7 and shCHRFAM7A cells, co-immunostained with SOX2, S100B, OLIG2 and NEUROD1 (arrowheads) at GW15 and GW17. Scale bar = 50 μm and 100 μm. **d** Quantification of SOX2, KI67, and EOMES upon knockdown of nAChRs. SOX2+ progenitors were reduced from 25% in

shCONTROL to 15.4% in shCHRNA7 ($p = 0.07$) and 14% in shCHRFAM7A ($p = 0.009$). KI67+ dividing progenitors were also reduced, with 8.3% in the presence of shCHRNA7 ($p = 0.03$) and 7.1% with shCHRFAM7A ($p = 0.01$), compared to 17% in the shCONTROL. **e** Quantification of NEUROD1+ immature neurons upon CHRFAM7A knockdown increased from 15.3% in shCONTROL to 30% in shCHRFAM7A ($p = 0.0032$). NEUN+ neurons upon nAChR knockdown showed the number of NEUN+ neurons increased from 22.5% in shCONTROL to 52% upon knockdown of CHRNA7 ($p = 0.009$), and to 53% upon knockdown of CHRFAM7A ($p = 0.01$). **f** Quantification of S100B+ astrocytes upon AChR knockdown reduced from 11 to 6% in shCHRNA7 ($p = 0.057$) and to 5.5% in shCHRFAM7A ($p = 0.05$). Quantification of OLIG2+ oligodendrocytes upon AChR knockdown reduced from 12% to 6% in shCHRNA7 ($p = 0.04$) and to 3% in shCHRFAM7A ($p = 0.01$), with $n = 3$. Source Data, two-sided t-test, *$p = 0.05$, **$p = 0.01$, ns not significant.

signal transducer shared by various cytokines including LIF. LIF signaling has recently been shown to regulate the production of cortical interneurons by oRG cells[55]. CALR is a chaperone protein involved in a variety of cellular processes including protein folding and calcium homeostasis, and CALR release recruits and activates microglia[56,57]. RCN2 (TCBP49) is a calcium-binding protein in the endoplasmic reticulum (ER), expressed in neurons and glia[58]. We also found that downregulated biological processes post CHRFAM7A knockdown included metabolic processes involving Oncostatin, LIFR, tyrosine kinase, and Wnt pathways (Supplementary Fig. 6e, f). We validated downstream targets of CHRNA7 and CHRFAM7A by RT-qPCR and found similar up or downregulation of UNC5C, BMP2, TKTL1, CXCL10, and HTR2A (Supplementary Fig. 6g) as observed in bulk RNA (Source Data). BMP2 encodes a TGF-β receptor ligand and is involved in critical developmental processes, including neurogenesis[59]. TKTL1 belongs to the transketolase family of enzymes and is preferentially expressed in neural progenitors[60]. A lysine-to-arginine substitution present in the modern human TKTL1 gene, but not present in the Neanderthal version, has been reported to increase the number of oRG cells in prenatal cortex and the neurons they produce[60]. CXCL10 is secreted by neurons and astrocytes and plays an essential role in cell migration[61–64]. HTR2A encodes a serotonin receptor expressed in RG, and mutations in this gene are associated with neuropsychiatric disorders including major depressive disorder and obsessive-compulsive disorder[65–67].

Comparing CHRNA7 and CHRFAM7A knockdown, we found shared candidates including CIR1, FBXL16, NHSL1 among upregulated and AK4P3, GSTM5, SCN1A among downregulated genes. (Supplementary Fig. 6h). Intriguingly, twice as many genes were differentially expressed between CHRNA7 and CHRFAM7A knockdown (Supplementary Fig. 6h, Source Data), suggesting that CHRNA7 and CHRFAM7A may mediate different downstream pathways in cortical cells. Our data highlight gene targets of nAChRs in the progenitor rich germinal zones of developing human cortex, including several genes that are implicated in neurodevelopmental and behavioral disorders.

## Cell type specific differential gene expression following nicotine exposure

To better elucidate the effects of nicotine exposure and nAChR subunit loss-of-function on specific cell types, we performed scRNAseq (Fig. 6a). We used dissociated primary cultures derived from second trimester tissue samples and treated them with nicotine to simulate nicotine exposure or performed shRNA-mediated knockdown of nAChRs (Fig. 6a). We examined two age groups, GW15-GW16 and GW19-22 and performed scRNAseq of 19,661 cells that passed quality control. GW15-GW16 corresponds to peak neurogenesis, while gliogenesis is ongoing at GW19-22[68]. Samples from GW15-16 will be referred to as neurogenic, while samples from GW19-22 will be referred to as gliogenic. Each sample showed a consistent distribution of known cortical cell types across ages (Supplementary Fig. 7a, b). UMAP clustering resolved the data into 25 clusters,

capturing all major cortical cell types and samples (Fig. 6b, c, Supplementary Fig. 7a, b).

To identify molecular changes downstream of nicotine exposure, we performed cell type specific and temporal DEG analyses (Source Data). Since the nAChR subunits, CHRNA7 and CHRFAM7A, were expressed by RG (Fig. 2a–c), we investigated neurogenic dividing RG in cluster 9. DEG analysis showed upregulation of genes involved in neurodevelopment, including NFIA, NFIB, HNRNPU, FNBP1L, NBEA, AUST2, and PTPRZ1 (Fig. 6d). NFIB is a transcription factor highly expressed in progenitors[69]. HNRNPU, an RNA-binding protein, is associated with neurodevelopmental disorders and seizures[70]. PTPRZ1 is a cell-surface protein enriched in oRG cells[71]. Genes downregulated after nicotine treatment included CELSR3, H3C3, CIC, RAB11B, AGRN, B4GALNT1, FLNA, L1CAM, NDRG4 and RUNDC3A (Fig. 6d). CELSR3 belongs to the cadherin receptor superfamily and flamingo subfamily, implicated in neuronal migration and axon guidance[72,73]. CIC is a transcriptional repressor regulating neural stem cell proliferation and neuronal differentiation[74,75].

To test nicotine exposure on early born excitatory neurons, we analyzed DEGs in cluster 0 post nicotine exposure (Source Data). Several genes, including NLGN4Y, ZEB2, CADM2, PBX1, PTBP2, SNHG14, CTNND2, were upregulated (Fig. 6e). PBX1 and PTBP2 act downstream of RNA-binding protein PTBP1 and have been associated with neuronal differentiation[76]. SNHG14 is a long non-coding RNA associated with the neurodevelopmental disorder Prader-Willi syndrome[77]. Several genes known to be functionally critical and highly expressed in the cortex were also downregulated such as DCHS1, SHANK3, LAMA5, NLGN2, LRP1, KIF1A (Fig. 6e). DCHS1 is a cadherin protein involved in establishing cortical architecture[78]. SHANK3 is involved in synapse formation and associated with ASD[79]. Similar DEG analyses of gliogenic dividing RG identified chromatin regulators such as PRR12, EHMT2, LENG8, and H1-10 to be upregulated post nicotine exposure (Source Data). EHMT2 is a potent regulator of gene transcription and methylates lysine 9 residues on histone H3 by recruiting epigenetic regulators[80]. GO analysis underlined semaphorin-plexin axon guidance signaling to be highly implicated downstream of nicotine exposure, in addition to processes such as cell adhesion, cell motility, and extracellular-matrix (ECM) organization (Fig. 6f). These findings emphasize that nicotine exposure alters a variety of genes and processes critical for cortical development in a temporal and cell type-specific manner.

We repeated our nicotine exposure experiments by exposing primary cortical cultures to low concentrations of nicotine (10 μM), approximating fetal exposure, for 5 days. We performed single-cell sequencing using the Split-seq method (Supplementary Fig. 8a) and collected approximately 20,000 cells from two biological replicates and their corresponding untreated controls. UMAP clustering revealed distinct, well-known cell clusters (Supplementary Fig. 8b), including RG, oRG, and various neuronal subtypes. All cell clusters expressed established marker genes (Supplementary Fig. 8c). DEG analysis

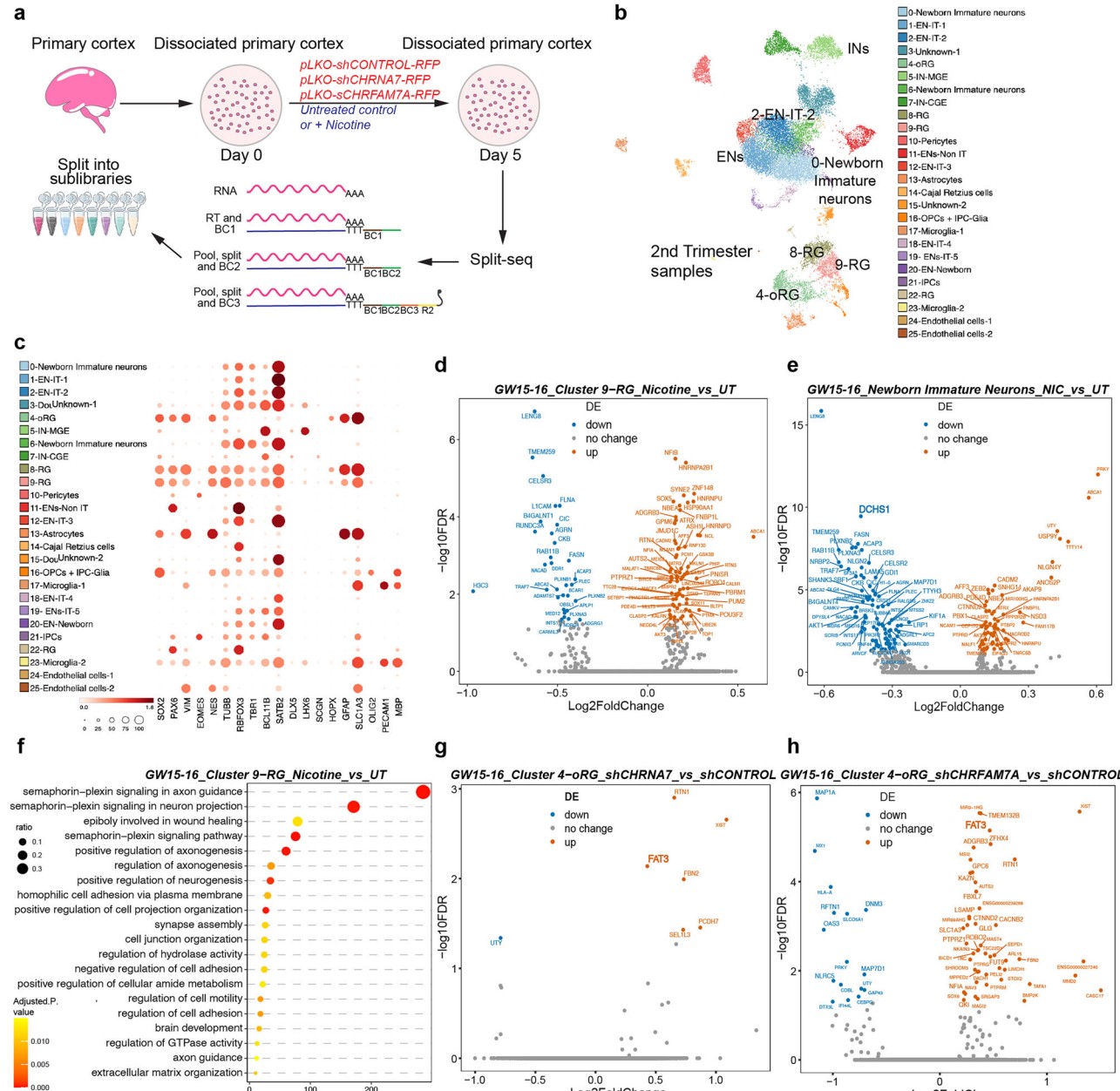

**Fig. 6 | Nicotine exposure and loss-of-function of AChRs alters Hippo signaling effectors. a** Workflow showing nicotine exposure and nAChR knockdown in primary dissociated cortical cell cultures. Cells were fixed on day 5 following the Parse Biosciences Fixation protocol and processed for Split-sequencing, a combinatorial barcoding approach where different barcodes are added in each step. We obtained 8 sublibraries. Gene expression libraries were prepared and sequenced using Illumina NovaSeq 2 Flowcell. **b** UMAP clustering of single-cell RNA sequencing (scRNAseq) data identifies major expected cell types. **c** scRNAseq analysis reveals anticipated patterns of marker gene expression across clusters. SOX2, PAX6, VIM, NES are expressed in radial glia and oRG clusters, while TUBB, RBFOX3, TBR1, BCL11B and SATB2 are expressed by neuronal clusters. **d** Genes up- and

downregulated upon nicotine exposure in dividing neurogenic RG cells. **e** Genes up- and downregulated upon nicotine exposure in immature neurons. One of the top downregulated genes is DCHS1, a canonical ligand of Hippo signaling. **f** Gene Ontology (GO) for dividing neurogenic RG identifies significant pathways and processes upregulated post nicotine exposure, including Semaphorin-plexin and homophilic cell adhesion signaling. **g** Genes up- and downregulated following knockdown of CHRNA7 in oRGs identifiies Hippo signaling receptor FAT3 among upregulated genes. **h** Knockdown of CHRFAM7A in oRGs also identifies Hippo signaling receptor FAT3 among upregulated genes. All DEGs are listed in Source Data.

identified several key genes, such as effectors from the Semaphorin-Plexin pathway and the Hippo ligand *DCHS1*. These changes in RG and Layer 5/6 excitatory neurons (EN) align with our findings using higher concentrations (Fig. 6e, f, Supplementary Fig. 8d–g). Gene ontology analysis highlighted Semaphorin-Plexin signaling, cell adhesion, and cytoskeletal organization in RG, while Layer 5/6 excitatory neurons showed alterations in post-synaptic density, microtubule organization, and synapse development (Supplementary Fig. 8e, g). Both high and

low nicotine concentrations resulted in consistent effects on RG and EN, including changes in the Hippo signaling effector, DCHS1.

## Cell type specific differential gene expression following nAChR loss-of-function

To investigate molecular changes upon loss-of-function of specific nAChR subunits, we identified cell-type specific DEGs and associated GO processes after shRNA-mediated knockdown of CHRNA7 and

CHRFAM7A. Knockdown of CHRNA7 in neurogenic RG produced upregulation of EPHA6, ZEB2, DAB1, and MYT1L, and downregulation of LAMA1, UBA7, and PLEKHM1 (Source Data). DAB1 protein plays a crucial role in reelin signaling during brain development[81]. MYT1L plays a key role in neuronal differentiation by specifically repressing non-neuronal genes[82]. LAMA1 mediates cell attachment, organization, and migration[83]. The associated GO processes for these DEGs include lysosomal transport, neuron fate specification, axon guidance, and differentiation (Source Data). The temporal shifts likely highlight differences in downstream molecular pathways between neurogenic and gliogenic RG mediated by CHRNA7 and CHRFAM7A (Supplementary Fig. 9a–e).

Knockdown of CHRNA7 induced an upregulation of FAT3, RTN1, FBN2, SEL1L3, PCDH7 in oRGs (Fig. 6g), and associated processes included histone H3-K27 and lysine demethylation and negative regulation of TGFβ signaling (Supplementary Fig. 9b). Knockdown of CHRFAM7A in neurogenic oRG led to an upregulation of genes including FAT3, ROBO2, CACNB2, KAZN, STOX1, PTPRM, FUT9, FAT3, ADGRB3, CASC17, FBXL7, and downregulation of DNM3, GAP43, MAP7D1, DTX3L, NLRC5 (Fig. 6h). FAT3, a Hippo signaling receptor, is a protocadherin involved in cortical development. Defects in FAT3 genes are implicated in schizophrenia[84]. ADGRB3 is highly expressed in the brain and enriched in the postsynaptic density (PSD), suggesting a role in synaptic signaling[85–87]. Other downregulated genes relate to cytoskeletal organization, for example, GAP43 is highly expressed during brain development and involved in neurite and synapse formation[88,89]. MAP7D1 is a microtubule associated protein involved in microtubule stabilization and colossal axon elongation[90,91]. Other associated GO processes include postsynaptic endocytosis, presynaptic vesicle budding, and dynamin family genes involved in migration (Source Data).

Knockdown of CHRFAM7A altered gene expression in GW19-22 Layer 5_6 ENs related to processes such as pre- and post-synaptic membrane assembly, genetic imprinting, histone modification and chromatin organization (Supplementary Fig. 9c, d). In late, upper layer 2_3 ENs, processes affected by CHRNA7 knockdown included generic neuron related functions such as synaptic membrane assembly, endocytosis, and cell-cell adhesion (Supplementary Fig. 9e). BMP and TGF signaling were also among the processes affected post knockdown of CHRNA7, while loss of CHRFAM7A changed sterol transport and PDGFRB pathways (Supplementary Fig. 9e). Overall, we identified temporal and cell type specific candidate genes and molecular processes post nAChR subunit manipulation, highlighting an ensemble of candidate genes uniquely regulated by these receptors.

## Hippo signaling effector YAP1 is downstream of Cholinergic signaling

Our DEG analyses identified several candidate genes that could be driving the downstream phenotypic changes observed in our agonist and knockdown experiments. Notably, we found that nicotine exposure downregulated the Hippo ligand DCHS1, while knockdown of both AChRs led to an upregulation of the Hippo receptor FAT3 (Fig. 6e, g, h). Yes-associated protein (YAP1), a co-activator in the Hippo pathway, is a downstream effector known to regulate cell fate. Gain-of-function of Yap1 expands the stem cell pool and reduces neuronal populations[92–94], while Yap1 loss promotes neuronal differentiation[94]. Nicotine exposure induces YAP1 expression and translocation to the nucleus[95]. To explore this potential signaling crosstalk and impact on YAP1 protein, we treated cortical cultures from GW16-19 ($n = 3$ biological replicates) with agonists or shRNAs targeting CHRNA7, CHRFAM7A, or a scrambled control. Protein lysates were collected on Day 5, and examined by Western blotting (Fig. 7a, b). We observed an increase in total YAP1 protein upon nicotine exposure and a reduction following knockdown of CHRNA7 and CHRFAM7A (Fig. 7b).

In silico analysis of published single-cell RNA-seq datasets revealed distinct expression patterns of Hippo signaling components across cortical cell types[96] (Fig. 7c). FAT3, CRB2 (Hippo receptors), and YAP1 RNA were highly expressed in RG and oRG cells, whereas DCHS1 was ubiquitously expressed across all cell types (Fig. 7c). Immunostaining confirmed the expression of DCHS1 and YAP1 proteins in the developing cortex (Fig. 7d). DCHS1 was enriched in SOX2+ and EOMES+ cells in the VZ and inner subventricular zone (iSVZ), with concentrated expression along apical endfeet, but was absent from the oSVZ and CP. YAP1 expression also overlapped with SOX2+ and EOMES+ cells in the VZ/iSVZ/oSVZ, but was undetectable in the CP. These findings support the progenitor-specific expression of YAP1 observed in our RNA-seq data (Fig. 7d).

To further investigate the impact of cholinergic signaling on YAP1 and to explore potential rescue, we treated organotypic cortical slices with the YAP1 inhibitor TAT-PDHPS1 in combination with ACh and nicotine. TAT-PDHPS1 (TAT) is a peptide inhibitor that consists of a cell-penetrating peptide (TAT) and an endogenous peptide (PDHPS1) known to inhibit YAP1 (Fig. 7e). We compared the phenotypic changes in treated samples with ACh, nicotine, and TAT treatment as well as untreated (UT) controls. Both ACh and nicotine exposure resulted in upregulation of YAP1 expression, with increased nuclear translocation of YAP1. TAT treatment induced cytoplasmic retention of YAP1, as expected. When combined with ACh or nicotine, TAT reduced YAP1 expression and nuclear localization (Fig. 7f). Quantification of HOPX+ oRG cells in the oSVZ relative to total DAPI-stained cells revealed a significant increase with both ACh and nicotine exposure. A complete rescue of oRG cell numbers was observed in ACh+TAT-treated samples, while nicotine+TAT treatment resulted in partial rescue, suggesting potential crosstalk between cholinergic and YAP1 signaling (Fig. 7h). In contrast, immunostaining for YAP1 revealed a significant loss of YAP1 protein in RFP+ cells following CHRNA7 and CHRFAM7A KD (Fig. 7h). This corroborates our previous findings showing a reduction in YAP1 expression following AChR KD (Fig. 7b).

Based on these observations, we propose a model of crosstalk between cholinergic signaling and the Hippo effector YAP1. Upon cholinergic activation, YAP1 translocates to the nucleus, where it interacts with transcription factors to regulate genes that balance stem cell maintenance and neurogenesis. Activation by ACh and nicotine enhances YAP1 expression and promotes its nuclear translocation, leading to an increase in RG at the expense of neurons. In contrast, nAChR knockdown reduces YAP1 expression and nuclear localization, shifting the balance toward neuronal differentiation and away from RG maintenance (Supplementary Fig. 10)[97]. Additionally, our loss-of-function experiments revealed a reduction in S100B+ astrocytes and OLIG2+ oligodendrocytes. These findings highlight the essential yet underexplored role of cholinergic signaling in modulating Hippo pathway activity, influencing the fate of RG, oRG, and glial cells, with potential implications for cortical development and disease.

## Discussion

The cerebral cortex in humans is composed of billions of functionally distinct neurons that are formed in a highly ordered manner during cortical development. Tightly regulated signaling cues are needed to determine the diverse fates of progenitor cells and their progeny. In this study, we identify a role of nicotinic receptor signaling in regulating proliferation and neurogenesis of RG and oRG cells in developing human cortex. We also use primary developing cortex and cortical organoids to characterize the expression of the nAChR subunit, CHRNA7, which is broadly expressed in mammalian brain, and CHRFAM7A, which is a human-specific gene arising from partial duplication of CHRNA7[24]. We used human ex vivo organotypic slices and organoids to model prenatal nicotine exposure. While these systems offer valuable insights, they do have limitations. For example, the duration of drug exposure was limited by the viability of our organotypic slices, and non-cortical brain regions that may contribute to clinical phenotypes were not represented in our organoids.

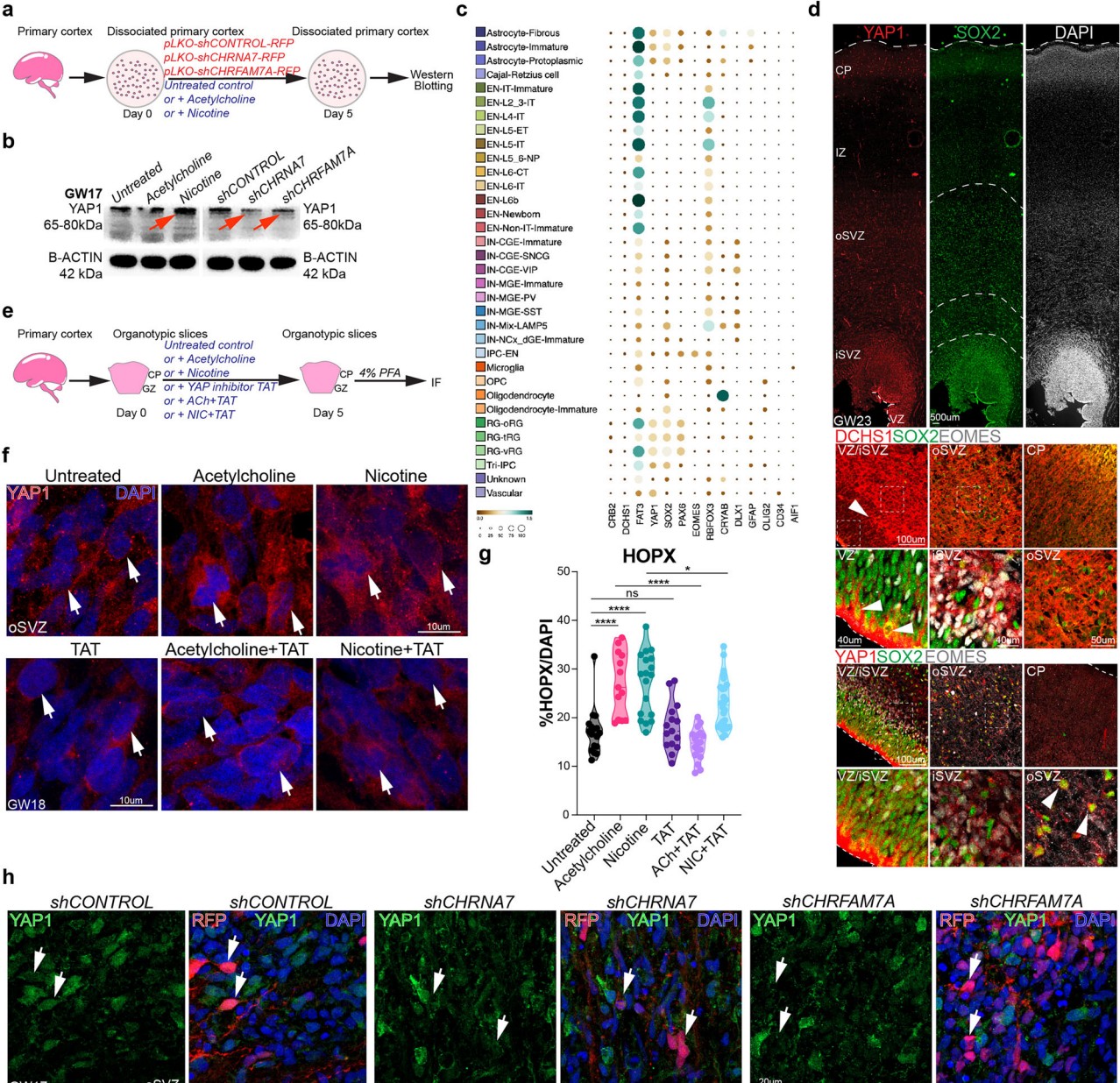

**Fig. 7 | Hippo signaling effector YAP1 is downstream of cholinergic signaling.** **a** Scheme showing the ACh, NIC exposure and nAChR knockdown in primary dissociated neural cultures, derived from whole cortices. Cells were treated from Day 0 to Day 5 and protein lysates were prepared RIPA buffer. **b** Western blotting for Hippo co-activator YAP1 showed an upregulation with NIC treatment and a downregulation with KD of CHRNA7 and CHRFAM7A, compared to respective control samples. B-ACTIN was used as a loading control. **c** In-silico analysis of published single-cell sequencing data shows the RNA expression of several known Hippo effectors in relevant neuronal cell types. YAP1, FAT3, CRB2 and DCHS1 are highly expressed in RGs and oRGs[96]. **d** Representative images showing protein expression of YAP1, DCHS1, SOX2 and EOMES in primary cortical tissue at GW23, $n = 3$. **e** Scheme showing the rescue experiment, where cortical organotypic slices were either treated with broad (ACh, NIC) agonists or a YAP1 inhibitor

(TAT- PDHPS1) or a combination of both, for 5 consecutive days. Samples were fixed with 4% PFA and processed for immunofluorescence. **f** Immunostaining shows upregulation of YAP1 protein expression and increased nuclear translocation (arrows) post agonist treatment in oSVZ, compared to untreated controls. TAT, Acetylcholine+TAT and Nicotine+TAT show relatively similar YAP1 expression and localization as the untreated control, $n = 3$. **g** Quantifications post ACh, NIC, TAT, ACh+TAT and NIC + TAT samples compared to untreated controls show full rescue of HOPX+ cells over total DAPI in ACh+TAT-treated, and partial rescue in NIC + TAT-treated samples (Source Data), two-sided t-test, *$p = 0.05$, **$p = 0.01$, ns not significant, $n = 3$. **h** Representative immunostaining for RFP, YAP1 and DAPI show downregulation of YAP1 protein in RFP+ cells in shCHRNA7 and shCHRFAM7A samples (arrows), compared to RFP+ cells in shCONTROL. (All scale bars are mentioned on the image panels).

CHRNA7 receptor subunits assemble to form pentameric α7 nAChRs. Previous studies have demonstrated that the expression of α7 nAChR increases in the rodent brain until postnatal week 1, after which it gradually decreases[13]. In the developing human cortex, nAChRs, including the α7 receptor, are present during the first trimester and increase in expression during later prenatal development[98,99].

CHRFAM7A codes for the peptide subunit dupα7, that can assemble with the native α7 subunit and negatively regulate receptor function in oocytes[27,28]. Despite previous research on the function of CHRFAM7A, its expression and role in the developing cortex remains unclear. We found robust expression of both nAChR subunits in SOX2 + RG progenitors in the VZ and oSVZ and observed enrichment of CHRFAM7A

in RG apical endfeet, hinting at potential signaling cues from the CSF. Interestingly, choline, a selective α7 activator, is present in high concentrations in the developing human brain[13,39,41] and we speculate it might be a source of cholinergic signaling in CSF. Another source of cholinergic signaling could be the cholinergic fibers that course through the oSVZ at around GW15, the IZ at around GW20, and the CP at later developmental stages. Overall, our results characterize the expression of CHRNA7 and CHRFAM7A in the progenitor niches of the developing human cortex.

NAChRs regulate neural maturation and synaptic transmission[13,24], but their potential role in progenitor cells during human cortical development is unclear. We show that activation of nAChRs with ACh and nicotine results in a significant increase in the number of proliferating progenitor cells in primary organotypic cortical slices and cortical organoids. The increase in progenitor cells is linked to a shift from neurogenesis to increased progenitor cell self-renewal. The same effects were seen with treatment of the α7 specific agonist PNU. Knockdown of the nAChR subunits CHRNA7 and CHRFAM7A led to inverse phenotypes compared to activation of nAChRs, supporting a role for both subunits in progenitor cell maintenance and neurogenesis.

Using bulk and single-nucleus RNA-sequencing, we obtained a more detailed understanding of the genes and processes affected by specific nicotinic cholinergic receptor activation in RG and oRG cells. The nAChR affected genes and GO processes involve cortical development and neurodevelopmental disorders, highlighting the significance of cholinergic balance in the developing progenitor niches, with potential clinical implications. Loss-of-function of CHRNA7 in neurogenic RG upregulated neuronal migration and differentiation genes including FAT3, DAB1, MYTL1 and LAMA1[81–83,100,101]. Knockdown of CHRFAM7A downregulated EGFR, CRB2, and FAT3, receptors involved in EGFR and Hippo signaling, key pathways known to play roles in cell proliferation, migration, and differentiation[102,103] underscoring the role of CHRFAM7A in stem cell maintenance and neuronal differentiation. CHRFAM7A is formed from the partial duplication of CHRNA7 and appears to have acquired a differential role during human brain development. Our results suggest that cholinergic signaling cues are required to maintain balance between stem cell maintenance and neurogenesis during human cortical development.

A previous study found higher expression of CHRFAM7A and CHRFAM7A/CHRNA7 RNA ratios in prenatal compared to postnatal prefrontal cortex[30] and also found elevated levels of CHRFAM7A expression and reduced levels of CHRNA7 in postmortem brains of patients with schizophrenia[24,30]. Previous organoid studies have shown morphological abnormalities, including reduced size of neural progenitor-rich rosettes, altered proliferation, and reduction in neuron numbers in organoids derived from schizophrenia patients compared to healthy controls[104,105]. Based on our results showing a functional role of CHRFAM7A in proliferation and neurogenesis of cortical progenitors, we speculate that CHRFAM7A might play a role in the morphological abnormalities observed in patient organoids. Though CHRFAM7A is formed from the partial duplication of CHRNA7, it appears to have acquired a differential role during human cortical evolution. Since nAChRs are expressed on multiple cell types during development, and they have implications for disease pathology, it will be important to further study the role of specific nicotinic receptor subunits in other cell types in the developing human brain.

TKTL1 was among the genes downregulated by CHRFAM7A knockdown. Modern human TKTL1 differs from the archaic neanderthal gene by a single amino acid substitution[60]. Overexpression of the modern human TKTL1, but not the Neanderthal TKTL1, in developing mouse and ferret neocortex results in an increase in oRG cells and a presumed eventual increase in neuron number[60]. In our results, knockdown of the human-specific nAChR subunit, CHRFAM7A, caused

downregulation of TKTL1, suggesting an evolutionarily important crosstalk. We identified twice as many exclusive DEGs following knockdown of CHRFAM7A (Source Data). This strengthens our hypothesis that CHRFAM7A has acquired human-specific biological function.

The exposure of the developing fetus to nicotine through maternal smoking is known to lead to alterations in brain morphology, including reduced cortical gray matter volume and cortical thinning, as well as behavioral changes[6,7,9–12]. In this study we show that nicotine causes a shift from neurogenesis to progenitor self-renewal in primary human cortical tissue and organoids. Although the long-term consequences have not been explored, we speculate that a persistent shift away from neurogenesis, could underlie decreased cortical volume and cortical thinning in the offspring of smokers. Using scRNAseq, we identified DEGs following nicotine exposure that could provide potential mechanisms for the altered cortical development induced by maternal smoking. Our analyses identified Hippo signaling effectors including FAT3, CRB2, DCHS1 and YAP1, to be downstream of nicotine-mediated cholinergic activation. We observed an increase in YAP1 protein levels following nicotine exposure and a reduction in YAP1 with AChR knockdown, along with corresponding changes in protein translocation. Mutations in Hippo signaling effectors FAT2 and DCHS1 have been linked to Val-Maldergem syndrome, which is characterized by periventricular heterotopias, cortical thinning, neuronal loss, and impaired neuronal migration[92,94], partially overlapping with the reported effects of maternal smoking. Our findings reveal a previously unexplored signaling interaction between the cholinergic and Hippo pathways that can influence the fate of RG and oRG cells, with potential pathological implications.

We identified several other genes affected by nicotine exposure that are linked to neurodevelopmental diseases. For example, HNRNPU, upregulated by nicotine exposure, is involved in developmental and epileptic encephalopathy[106] and AUTS2, also upregulated after nicotine exposure, is associated with ASD and schizophrenia[107–109]. We also identified semaphorins and plexins to be affected after nicotine exposure (Fig. 6f, Supplementary Fig. 8e). Semaphorins and plexins are key regulators of cell proliferation, axon guidance, neuronal migration, laminar segregation, central nervous system patterning, and glial recruitment, all critical processes in cortical development[110]. We also observed an upregulation of chromatin regulators in late stage dividing RG. GO analyses highlighted processes such as chromatin silencing and organization, suggesting a vulnerability to chromatin-related dysregulation post nicotine exposure (Source Data). Our study aimed to identify selective transcriptional changes and potential vulnerabilities of specific cell types to prenatal nicotine exposure. However, we only investigated early and late second trimester cortical development. Future studies should address effects at later stages of cortical development.

## Methods
### Lead contact and materials availability
The research adheres to all applicable ethical guidelines and has received approval from UCSF under the GESCR protocol. Any questions and requests for resources should be directed to Arnold Kriegstein, at Arnold.Kriegstein@ucsf.edu.

**Pluripotent stem cell lines.** The following pluripotent stem cell lines were used: H1/WA01 (Human embryonic stem cell line, WiCell, RRID: CVCL_9771, Male); WTC-11 human iPSC (Conklin laboratory, Gladstone Institutes, GM25256, Male) and 1323-4 human iPSC (Conklin laboratory, Gladstone Institutes, RRID:CVCL_0G84, Female).

**Primary human cortex tissue.** Primary human cortical tissue was obtained and processed as approved by UCSF Gamete, Embryo and Stem Cell Research Committee (GESCR). All samples were de-identified with no information on sex. Tissue was collected with

patient consent for research and in strict observance of legal, institutional, and ethical regulations.

**Cortical organoid differentiation protocol.** Three PSCs (H1, 1323-4 and WTC11) were differentiated into cortical organoids based on a published protocol for directed forebrain differentiation[42,111]. PSCs were dissociated into single cells using accutase and plated into 96-well ultra-low attachment V-bottom plates at a density of about 10,000 cells per well. Cells were aggregated in GMEM (Lifetech; #11710-035) containing 20% v/v Knockout Serum Replacement (KSR; Lifetech; #10828-028), 1x Penicillin-Streptomycin (PS; Gibco), 1x non-essential amino acids (Gibco), 11 mg/ml Sodium Pyruvate (Thermo Fisher; #11360070) and 0.1 mM 2-Mercaptoethanol (Sigma). Fresh Rock inhibitor Y-27632 (20 μM; Tocris; #1254), TGF-B inhibitor SB431542 (5 μM; Tocris; #1614) and WNT inhibitor IWR1-endo (3 μM; Caymen Chem; #13659) were added at every medium change until day 6, 18, and 18, respectively. Medium was changed every other day. At day 18, cell aggregates were transferred to low adhesion 6-well plates and moved to an orbital shaker rotating at 90 rpm. The culture medium was changed to DMEM/F-12 Glutamax (Gibco) containing 1X N2 supplement (Lifetech, #17502-048), 1X CD Lipid concentrate (Lifetech; #11905-031), 1X PS and 2.5 μg/ml Amphotericin B (Gibco). At day 35, the medium was changed to DMEM/F-12 Glutamax containing 1X N2, 1X CD Lipid concentrate, 1x PS, 10% v/v Fetal bovine serum (FBS; HyClone; #SH30071.03), 5 μg/ml Heparin (Sigma; #H3149), 0.5% v/v growth-factor reduced Matrigel and 2.5 μg/ml Amphotericin B. From day 70 onwards, cerebral organoids were maintained in DMEM/F-12 Glutamax, 1X N2 supplement, 1X CD Lipid concentrate, 1X PS, 10% v/v FBS, 5 μg/ml Heparin, 1% v/v growth-factor reduced Matrigel, 1X B27 supplement (Lifetech; #17504-044) and 2.5 μg/ml Amphotericin B. All three PSC lines were utilized for treatment experiments in cortical organoids.

**Dissociation cultures.** Primary cortical tissue was cut into small pieces and dissociated using the Papain Dissociation system (Worthington). Primary tissue was incubated in Papain (Worthington; #LK003178) resuspended in Earle's Balanced Salt Solution (EBSS) and containing DNAse (Worthington; #LK003170) at 37 °C for 30–40 min. Falcon tubes were inverted 10 times every 10 min. After incubation, samples were mechanically triturated using a P1000 pipette to obtain a single cell suspension and spun down at 300 g for 5 min. Supernatant was aspirated and cells were resuspended in DMEM/F-12 Glutamax containing 1X B27, 1X N2, 1X Sodium Pyruvate and 1X PS, filtered through 40 μm nylon mesh cell strainers (Fisherbrand) and counted. For qPCRs, 500,000 cells or 1 – 1.5 Mio. cells were plated for 12-well plates or 6-well plates, respectively.

**Organotypic slice cultures.** Primary human cortical tissue was obtained in artificial cerebrospinal fluid (ACSF) containing 125 mm NaCl, 2.5 mm KCl, 1 mm MgCl$_2$, 2 mm CaCl$_2$, 1.25 mm NaH$_2$PO$_4$, 25 mm NaHCO$_3$, 25 mm D-(+)-glucose. ACSF was bubbled with a 95% O$_2$–5% CO$_2$ gas mixture before use. Tissue was embedded in 3.5% w/v low melt agarose (Fisher Scientific; BP165-25) and sectioned into 300 μm slices using a vibratome (Leica). Slices were transferred to Millicell inserts in a 6-well plate and cultured at the air-liquid interface with slice culture medium (66% v/v Basal Medium Eagle, 25% v/v Hank's Balanced Salt solution, 5% v/v FBS, 0.67% w/v glucose, 1X Penicillin-Streptomycin-Glutamine, 1X N2 supplement). Medium was changed every three days. For ex vivo knockdown assays, organotypic cortical slices were infected with lentiviruses on day 0 and 4 and harvested for immunohistochemistry on day 7.

**Small molecule treatments.** Organotypic slices and iPSC-derived cerebral organoids were treated by daily addition of small molecules to the respective culture medium for 4 days (as depicted in respective experimental paradigms). Small molecules added were 10 μM and 500 μM (Tocris; #3546), 100 μM acetylcholine (ACh) (Tocris; #2809, 10 μM PNU-282987 (PNU) (Tocris; #2303) or 5 μg/ml a-bungarotoxin () (Tocris; #2133). PNU was dissolved in DMSO according to manufacturer's recommendation and all other small molecules were dissolved in H$_2$O, thus, as a control, organotypic-slices and organoids were DMSO-treated or untreated, respectively. YAP1 inhibitor, TAT-PDHPS1 (Tocris; #7832), 10umol/l was dissolved in dissolved in H$_2$O, following the manufacturer's recommendation.

**RNAscope and BaseScope.** The RNAscope Multiplex Fluorescent Reagent Kit v2 and BaseScope Detection Reagent Kit v2 – RED were purchased from ACD Advanced Cell Diagnostics. RNAscope was conducted for CHRNA7 and BaseScope for CHRFAM7A according to the manufacturer's recommendation. BaseScope is used to detect short target RNA sequences between 50 and 300 nucleotides. This technique is also used to detect exon junctions and splice variants. Probes for CHRNA7 (Hs-CHRNA7; 310101, targeting 181-1591 region) and CHRFAM7A (BA-Hs-CHRFAM7A-E4E5; 704391, targeting 617-658 region) were purchased from ACD biosciences. Opal 570 (Akoya Biosciences) was used as a fluorophore. After RNAscope or BaseScope, immunohistochemistry was conducted as described below but without a target retrieval step and using double the concentration of primary antibodies indicated in Table 2.

**Immunohistochemistry.** Primary cortical tissue samples were fixed in 4% PFA overnight at 4 °C, washed with 1x PBS and equilibrated in 30% sucrose in 1x PBS overnight at 4 °C. The samples were embedded in cryomolds in 3:2 O.C.T. (Tissue-tek): 30% sucrose solution, cryopreserved on dry ice and stored at −80 °C. Frozen tissue was sectioned into 16μm sections onto Fisherbrand™ Superfrost™ glass slides and stored at −80 °C. Antigen-retrieval was performed with 1X boiling citric acid-based antigen unmasking solution (Vector Laboratories) for 20 min. Slides were washed once with MilliQ-H$_2$O and twice with PBS and blocked in Donkey blocking buffer (DBB) containing 5% donkey serum, 2% gelatin and 0.1% Triton X-100 in PBS for 30 min at RT. Primary antibodies (see Table 2) were incubated in DBB overnight at 4 °C, washed three times with 0.1% Triton X-100 in PBS and incubated with AlexaFluor secondary antibodies (ThermoFisher and Jackson Labs) in DBB for 3 h at RT. Organotypic slice cultures were processed like

**Table 2 | Antibodies used for immunohistochemistry**

| Target Protein | Host species | Manufacturer | Identifier | Dilution |
|---|---|---|---|---|
| AChE | mouse | Abcam | ab2803 | 1:500 |
| CHRFAM7A | rabbit | Aviva system biology | OAAN02465 | 1:250 |
| CTIP2 | rat | Abcam | ab18465 | 1:500 |
| EOMES | sheep | R&D | AF6166 | 1:200 |
| HOPX | rabbit | Proteintech | 11419-1-AP | 1:250 |
| KI67 | rat | Abcam | ab156956 | 1:300 |
| NEUN | guinea pig | Millipore | ABN90 | 1:500 |
| SOX2 | mouse | Santa Cruz | Sc-365823 | 1:500 |
| SATB2 | mouse | Abcam | ab51502 | 1:500 |
| VIMENTIN | chicken | EMD Millipore | Ab5733 | 1:250 |
| ZO-1 | rabbit | Proteintech | 21773-1-AP | 1:300 |
| DLX5 | rabbit | Sigma | HPA005670 | 1:100 |
| YAP1 | rabbit | Novus Biologicals | NB110-58358SS | 1:300 |
| DCHS1 | rabbit | Novus Biologicals | NBP2-13901 | 1:300 |

primary cortical tissue, except that PFA fixation was conducted for 45 min at room temperature and slices were cryosectioned into 12 µm sections. Cortical organoids were fixed, embedded, sectioned, and stained like primary cortical tissue, except that they were fixed in 4% PFA for 45 min at room temperature, and processed into 14 µm sections. Images were acquired using a Confocal microscope (Leica TCS SP5 X).

**TUNEL staining.** Tissue slides were post-fixed with 4% PFA in TBS for 15–20 min at RT, washed in 1X TBS for 30 min and permeabilized using 0.1% Triton X-100 for 2 min. Slides were washed with TBS, each wash for 3 min, with 1X TBS. TUNEL mix from Roche In Situ Cell Death Detection Kit- FITC, composed of TUNEL label and enzyme were mixed in a proportion of 10:1, and added to slides for 1 h at 37 °C. Slides were washed 3X with TBS at RT for 5 min each, before mounting.

**Western blotting.** Protein lysates was prepared from primary tissue and organoids through trituration in RIPA buffer (Thermo Fisher; #89900) with complete protease inhibitor cocktail (Millipore-Sigma; #P8340-1ML). Samples were centrifuged at 12,000 g for 10 min and the supernatant was used while the pellet was discarded. A protein mixture of 50:25:20:5 protein sample: NuPAGE LDS Sample Buffer (Invitrogen; #NP0008): RIPA buffer: 2-mercaptoethanol was boiled at 95–98 °C for 10 min and separated using 4–12% Polyacrylamide gels (Invitrogen; #NP0335BOX) in 1x NuPAGE MES SDS Running Buffer (Invitrogen; #NP0002) at 110 V. Protein bands were transferred to Nitrocellulose membranes (Protan, GE) in MilliQ water containing 1x transfer buffer (Invitrogen; #NP00061) and 10% v/v methanol at 90 V at 4 °C for 1 h. Membranes were blocked in blocking buffer (5% w/v milk powder in 1x tris buffered saline with Tween 20 (TBS-T) (Santa Cruz; #sc-362311 diluted in MilliQ-H₂O) for 1 h at room temperature and incubated at 4 °C overnight in primary antibody diluted in blocking buffer (see Table 3). Membranes were washed three times for 15 min. in 1X TBS-T, incubated in secondary antibody (see Table 3) in blocking buffer for 2 h at room temperature and washed in 1X TBS-T three times for 15 min. Blots were detected using the SuperSignal™ West Pico PLUS Chemiluminescent Substrate (Thermo Scientific; #34579) in a Chemi-Doc MP UV-Vis Spectrometer (Bio-Rad).

**Pluripotent stem cell culture.** PSCs, frozen in mFreSR (Stem Cell Technologies; #5855), were thawed and cultured on growth-factor reduced Matrigel (BD Biosciences; #354230)-coated six well plates in Stem Flex Pro Medium (Gibco) with 10 µM Rock inhibitor Y-27632. Medium was changed every other day without Rock inhibitor until the cells reached about 85–90% confluency. For maintenance passaging, ReLeSR (Stem Cell Technologies; #100-0484) was added to cells for 1 min, aspirated, and cells were incubated at 37 °C for 5 min. Cells were resuspended in Stem Flex Pro Medium and plated into new growth-factor reduced Matrigel-covered plates.

**Quantification Strategy.** For small molecule experiments, immunohistochemistry was performed for known markers of radial glia, intermediate progenitors, proliferation, deep and upper layer neurons. Several sections across multiple biological replicates were imaged post immunohistochemistry. Quantifications were performed using IMARIS software and cell numbers were normalized by total number of DAPI per section. For knockdown experiments, the number of double positive cells in shCHRNA7 and shCHRFAM7A were normalized over total RFP positive (infected) cells and compared to shCONTROL. Each data point on every graph represents an image captured and quantified.

**Bulk RNA sequencing.** For bulk RNA samples, knockdown was performed in dissociated primary cultures using lentiviral-mediated shRNAs. Four biological (GW15, 18, 21, 22) and two technical replicates each were prepared, and RNA was isolated and purified as described for qPCR. For shRNAs and small molecule treatments, cells were harvested three- and four-days post treatments start, respectively. RNA samples were analyzed for their integrity and concentration using Agilent 2100 Bioanalyzer and only samples with RIN scores above 9.0 were processed for sequencing. Bulk RNA-sequencing was performed at Genewiz using 150 bp paired-end RNA-seq on an Illumina Hi-Seq platform. Reads were aligned and quantified using kallisto and quantified at gene level with tximport. Differential gene expression analysis was performed using limma/voom, accounting for $n = 2$ biological replicates per timepoint, with $n = 2$ technical replicates within each individual. P values were corrected for multiple comparisons using the Benjamini-Hochberg false-discovery rate adjustment (Source Data).

**Single-cell sequencing.** Primary cortical tissue corresponding to neurogenic-rich (GW15-18) and gliogenic-rich (GW19-23) time points were dissociated and treated either with NIC or shRNAs on consecutive days as above. Cells were harvested at day 5 and samples processed using the Parse Biosciences 100WTA kit according to the manufacturer's recommendation. This method involves a combinatorial and sequential barcoding of fixed cells or nuclei followed by generation of sublibraries. 8 sublibraries were prepared and tested with BioAnalyzer for quality. Sequencing was performed using Illumina Nova-Seq S2 flow cell and data analyzed using the Parse Computational pipelines.

**Bioinformatics analysis- Bulk-RNAseq.** Reads were aligned and quantified using kallisto and quantified at gene level with tximport. Differential gene expression analysis was performed using limma/voom, accounting for $n = 2$ biological replicates per timepoint, with $n = 2$ technical replicates within each individual. P values were corrected for multiple comparisons using the Benjamini-Hochberg false-discovery rate adjustment (Source Data).

**Bioinformatics analysis - scRNAseq.** Nova-Seq data from 8 sublibraries were mapped to GRCh38/hg38 and demultiplexed using split-

**Table 3 | Western blotting antibodies**

| Primary / Secondary | Target protein | Species | Manufacturer | Cat. # | dilution |
|---|---|---|---|---|---|
| Primary | GAPDH | mouse | Sigma | G8975 | 1:5000 |
| Primary | HISTONE H3 | mouse | Cell Signaling | 3638S | 1:1000 |
| Primary | B-ACTIN | rabbit | Cell Signaling | 4970S | 1:1000 |
| Primary | CHRNA7 | rabbit | Alomone labs | ANC-007 | 1:500 |
| Primary | CHRFAM7A | rabbit | Aviva system biology | OAAN02465 | 1:500 |
| Primary | CHRNB1 | rabbit | Proteintech | 11553-1-AP | 1:500 |
| Primary | CHRNB2 | rabbit | Abcam | ab41174 | 1:500 |
| Secondary | Anti-Mouse HRP | rabbit | Abcam | ab6728 | 1:10,000 |
| Secondary | Anti-Rabbit HRP | goat | Abcam | ab6721 | 1:10,000 |

pipe[112] (v1.1.1) from the parse biosciences pipeline with the following parameters: --mode all --chemistry v2 --kit WT. Then the outputs of each sublibrary were combined into a single dataset using split-pipe --mode comb. Gene expression matrices were loaded into the Python package Scanpy[113] (1.9.3) for quality control and downstream analyses. We removed cells based on these criteria: (1) number of UMIs ≤200, (2) number of UMIs ≥98-percentile, and (3) percentage of mitochondrial genes ≥20%. Cell doublets were detected and removed using scvi-tools (0.20.3)[114]. Additionally, genes expressed in less than 10 cells were discarded. Clean data were then normalized and transformed with Scanpy. Then we leveraged scvi-tools to correct batch effects. To cluster the cells and identify cell types, we used Scanpy for downstream analysis.

For differential genes analysis, FindMarkers from Seurat[115] (v4.3.0) was applied to compare two conditions. We combined GW15 and GW16 as 2nd_early time point, and GW19 and GW22 as 2nd_late time point. Genes with |log2FoldChange| > 0.25 and p.adjust < 0.05 were considered as differentially expressed genes. For GO terms enrichment, enrichR[116] (v3.2) and databases "GO_Molecular_Function_2021", "GO_Cellular_Component_2021", "GO_Biological_Process_2021" were used for functional enrichment.

**Quantifications and statistics.** Quantifications were conducted manually using IMARIS Software. For statistical analysis, an unpaired two-tailed student's t-test using prism software was used. For phenotypic analyses, to handle repeated measures and paired samples when analyzing staining data, a linear mixed effect model [lme4 lmerTest references] was used to statistically model the effect of agonist, antagonist, and shRNA knockdowns. Staining was calculated as a percentage of DAPI cells within the image, which generated an approximately normal distribution of values. Each staining was modeled as follows:

Percentage Positive ~ Condition + (1|Sample).

For a given protein stain, models were fit within each tissue type (Organoids week 4, Organoids Week 8, Slice cultures), with only relevant conditions. Untreated results was the background for results from ACh, NIC, and BTX treatments, DMSO for PNU, and shCONTROL for shChRNA7 and shCHRFAM7A.

lme4: https://www.jstatsoft.org/article/view/v067i01.
lmerTest: https://www.jstatsoft.org/article/view/v082i13.
dharma: https://cran.r-project.org/web/packages/DHARMa/vignettes/DHARMa.html.

TUNEL staining was not normally distributed, because cell death was low, it generated more empty values than expected by normal distribution. So instead, we took the mean for each sample/condition, and conducted a paired, 2-tailed t test.

**Reporting summary**
Further information on research design is available in the Nature Portfolio Reporting Summary linked to this article.

## Data availability
Original source data from the study is available in Source Data. RNA-seq data will be available in dbGAP upon publication at dbGAP study accession: phs000989.v7.p1. Source data are provided with this paper.

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

## Acknowledgements

The authors thank Dr. Arpana Arjun and members of the A.R.K. laboratory for useful discussions. This study was supported by the NIH award R35NS097305 to A.R.K., Swiss National Science Foundation Early Postdoctoral Mobility fellowship to T.M. and Boehringer Ingelheim Fonds PhD Fellowship to C.V.S.

## Author contributions

Author contributions: T.M. and M.P.P. conceived the project. T.M., M.P.P., C.V.S., M.K.H, J.L. and A.R.K. designed research; T.M., C.V.S., M.W., T.I.H., J.R., S.W., Q.B., J.B., V.U., M.S., L.Z., I.L.C., I.M., M.A.A. and L.Z. performed research; Y.W., T.M., C.V.S., M.P.P., T.W., G.Z., V.U., M.S., M.W. and J.B., analyzed data; C.V.S., T.M., Y.W., M.P.P. and A.R.K. wrote the paper; and T.M. and A.R.K. supervised.

## Competing interests

A.R.K. is a co-founder, consultant, and member of the Board for Neurona Therapeutics. The other authors declare no conflicting interests.
