## [Transparent Peer Review file · Nature Communications]

$\alpha 7$ nicotinic acetylcholine receptors regulate radial glia fate in the developing human cortex

Corresponding Author: Dr Tanzila Mukhtar

Version 0:

Reviewer comments:

Reviewer #1

(Remarks to the Author)

This paper investigates the effects prenatal nicotine exposure on human neuronal development. The authors used primary organotypic slices and organoid cortical cultures to investigate this question. The data obtained by the authors point at the participation of CHRNA7 and CHRFBAM7A nicotinic subunits in this process. These receptors use YAP1 for downstream signaling. The main results are that nAChR activation disbalances the ratio between neural progenitors and neurons by decreasing the neuronal population in the developing cortex. In line, loss-of-function experiments using sh-RNAs induced inverse effects compared to those promoted by CHRNA7 and CHRFBAM7A activation. Knock-down of CHRNA7 and CHRFBAM7A results in upregulation of YAP1 expression.

The manuscript uses an excessive number of methods, including knock-down and pharmacological treatments, RNA scope, bulk and single cell RNAseq and bioinformatics analysis. Statistical analysis is adequate. The work is significant and meets the expected standards for publication.

Some points, which might be useful:

1. The significance of the work is clear for human health, as premature nicotine exposure affects cortical development. However, how the chosen nicotine dose reflects conditions of nicotine concentration in the premature brain?
2. Nicotine is a selective nAChR agonist, while acetylcholine is also a mAChR agonist. Further, nicotine is more stable than acetylcholine in differentiation assays. I do not understand why acetylcholine was used instead of choline, which would activate alpha7 receptors. Further, a functional calcium imaging assay, confirming the functionality of these receptors and possible differences between cell responses would be helpful. Anyway, which nicotine concentration was used for the studies
3. Astrocytes, the most abundant glial cell type in the brain, are underrepresented in traditional cortical organoid models due to the delayed onset of cortical gliogenesis. In view of that, astrocytes in cortical development was not assessed in this paper.
4. Authors state that they use iPSC-derived cortical organoids. In the Methods' section two iPSC cell lines (one male and one female) and a human embryonic stem cell line. However, i.e. Figure legend 2 does not state which cell line was used. Were any differences obtained between male and female iPSC-derived progenitors?
5. Authors performed multiplexing (simultaneous staining) for Ki-67 expression with cellular phenotyping and CHRFBAM7A expression? This is not stated in the Methods' section. This can be done by imaging cytometry and much more easily using state-of-the-art flow cytometers.
6. The Western blots, especially for CHRNB1 and CHRNB1, in Extended Data Fig. 2 are not acceptable. Which negative control was used to probe for the selectivity of the antibody? Why are the staining intensities for the GAPDH control different, and why GAPDH shows a double band in one occasion (d) and a single band in (e) and (f).
7. The authors need to comment on the limits of their work, compared to in vivo conditions.
8. There are some inaccuracies in the writing style, please, check i.e. "Activation of nAChRs increases progenitors and decreases neurons in developing human cortex". "NIC is known to primarily act on nAChRs while ACh also acts on muscarinic acetylcholine receptors." There is also an excess of abbreviations. Why nicotine needs to be abbreviated as "NIC"?

Reviewer #2

(Remarks to the Author)

In this work, Mukhtar et al., studied the role of $\alpha 7$ nicotinic acetylcholine receptors in regulating cell fate in the developing human cortex. They focused on two nAChR subunits, CHRNA7 and CHR FAM7A, expressed in both progenitors and neurons of fetal cortex. They modeled prenatal exposure to nicotine (NIC) by activating these receptors pharmacologically and studied their role during human cortical development by protein quantification, bulk and scRNA-seq in both fetal cortices' slices and cortical cell cultures. Their findings were corroborated by nAChR knockdown experiments. The study demonstrated that activation of these receptors results in an increase of progenitor cells and radial glia and a decrease of both deep and upper-layer neurons.

Although previous studies on prenatal NIC have been conducted in rodents and non-human primates, this study is unique in its exploration of these mechanisms in human fetal brain specimens. I believe it holds significant importance for the neurodevelopmental field, particularly because it focuses on two nAChRs previously associated with neurological disorders like SCZ and BD. Overall, the study is of great interest, proposing a novel signaling interaction between Hippo pathway and cholinergic signaling, though improvement on a few points would make it an outstanding study.

Major comments

The authors investigated CHR FAM7A gene and protein expression in both ex vivo and organoid slices but did not do so for CHRNA7 (Figure 2). Which antibody was used—was it the same one used for the western blot assay? Why was immunofluorescence for CHRNA7 not included?

Regarding protein analysis by western blot, could the authors comment on the presence of additional bands in Ex. Fig. 2c-f? Are these known isoforms? In Ex. Fig. 2c-d, additional bands of both higher and lower molecular weights are observed in some samples. Were these antibodies previously validated?

The molecular weight of CHR FAM7A is unclear, as it appears to be 50 kDa in panel c but 55 kDa in panel d. Please verify and correct this discrepancy. Also, in Ex. Figure 2f, a completely different molecular weight (100 kDa) is reported for CHR FAM7A in the organoid lysate. Why does this discrepancy exist? Please check and clarify.

In the manuscript, page 4, the authors claim higher expression of CHR FAM7A in the germinal zone. Quantification of the protein bands from the germinal lysates would strengthen this claim.

The pharmacological functional studies of the cholinergic system were elegantly designed, with appropriate controls and a TUNEL assay to exclude treatment-induced cell death or apoptosis. However, few points require clarification:

- What criteria or references were used to determine the concentrations of the small molecules in the study?
- Can the authors comment on the EOMES results? While both molecules increased progenitor proliferation (SOX2+) and downregulated neuronal markers (NEUN, CTIP2, and SATB2), why did NIC uniquely favor the EOMES fate?
- NIC exposure has previously been associated with GABAergic alterations. Besides DLX5 quantification (Ex. Fig. 3), considering that the authors focused only on cortical samples, this model lacks the major interneuron source, the basal ganglia. Given this limitation, the authors should be cautious in drawing conclusions regarding NIC's effects on this cell type, which was not fully investigated.

The data involving the antagonist BTX in combination with NIC and ACh are less convincing and suggest that ACh and NIC actions are not exclusively mediated by the nAChRs studied. The authors themselves mention this on page 7. This limitation should be explicitly addressed in the discussion.

Do NIC and acetylcholine treatments alter the expression of different nAChRs in cortices and/or organoids?

The protein distribution of AChE in cortical tissue (Fig. 1) was well studied. Do cortical organoids express AChE?

Did the authors investigate the long-term effects of this initial increase in progenitors? Does this initial insult eventually lead to an increase in neurons, or is the aberrant overproduction of progenitors compensated for over time?

How was the 4-days exposure paradigm chosen?

RNA bulk and scRNA sequencing data are interesting, highlighting several targets of the nAChR, but the text and the supporting figures need some editing. To mention a few examples, the heatmap Ext.Fig 5c gene labels are too small; the gene CACNA1A mentioned by the author as upregulated by shCHRNA7 is hard to be found in any Table/Data and if in the heatmap it is not visible. Only found in the qPCR validation tab and Ext.Fig.5g. Please provide better panels in the figure that can show these findings.

The investigation of molecular changes after NIC exposure by scRNA-seq, per cell types corroborate the previous finding of its role in cortical development followed by the discovery of inter-talk between nAChRs and YAP pathway make the paper very important, highlighting the necessity of further study for developmental disorders of these receptors, also in other cell types.

Minor comments

Authors might want to consider including a panel with the experimental design, tissue and organoids developmental stages analyzed in Figure 1.

In Extended Figure 2d, the authors could add a small legend clarifying the distinction between "germinal zone lysate" and "whole cortices lysate" shown in panel c.

In Figure 4b, the brightness of CTIP2 panels should be increased to improve the visualization of CTIP2+ cells or, alternative, the channel color could be changed to gray instead of yellow. The same applies to Extended Figure 4b.

In the Loss-of Function nAChRs experiments using shRNAs (page 7), the authors should clarify at the beginning of the paragraph that the assay was first performed on dissociated cortical culture and then on organotypic slices. The EOMES+ results (line 273) are described as consistent with previous observations. However, I disagree, as NIC treatment increases the EOMES population, while PNU shows no change. Please verify this claim.

In the discussion (page 14), authors claim to have identified twice as many exclusive DEGs for CHRFA7A knockdown. Please curate the Tables and reference a corresponding Figure (Fig.6g-h?).

A legend tab is highly recommended for all Tables.

The Material and Methods section is well written, though under Small molecule treatments the concentration of the first molecule listed (NIC) is missing.

Reviewer #3

(Remarks to the Author)

Prenatal nicotine exposure has been associated with a reduction in fetal cortical gray matter volume. To elucidate the underlying mechanisms, researchers investigated the role of two nicotinic acetylcholine receptor (nAChR) subunits, CHRNA7 and the human-specific CHRFA7A, in human cortical progenitors and neurons. Their findings identified a novel function of nicotinic receptor signaling in regulating the proliferation, self-renewal, and neurogenesis of radial glial (RG) and outer radial glial (oRG) cells. More specifically, pharmacological activation of nAChRs, either through broad-spectrum or specific agonists, resulted in an increased number of RG cells while concomitantly reducing the populations of deep-layer and upper-layer neurons. Conversely, knockdown of nAChRs led to a reduction in RG cells and a corresponding increase in neuronal populations, though the precise neuronal subtypes affected remain to be determined. These findings indicate that CHRNA7 and CHRFA7A play a crucial role in mediating fate transitions in response to prenatal nicotine exposure. scRNAseq analysis further revealed that nicotine exposure downregulates key genes essential for excitatory neuron function. Notably, many of these transcriptional alterations were uniquely regulated by either CHRNA7 or CHRFA7A, suggesting an evolutionary divergence in their regulatory functions. Further investigation identified the Hippo signaling co-activator YAP1 as a downstream effector of nAChR activation, thereby establishing a mechanistic link between prenatal nicotine exposure and alterations in RG and oRG cell populations through YAP1-mediated signaling. These findings provide critical insights into how prenatal nicotine exposure disrupts cortical development by influencing progenitor cell fate decisions and excitatory neuron gene expression. By distinguishing the distinct roles of CHRNA7 and CHRFA7A, the study highlights an evolutionary divergence in cholinergic regulation of neurodevelopment. Collectively, the dataset and conclusions presented are scientifically robust and contribute to a deeper understanding of the impact of nicotine on the developing brain.

Major comments

Electrophysiological analysis of nAChR function is missing. What are the membrane effects of activating the $\alpha 7$ homomeric nAChR and the human-specific variant formed by the CHRFA7A-encoded subunit? Do they induce depolarization in RG and oRG cells, thereby activating VGCCs and triggering calcium-dependent downstream molecular signaling? This would also be important to support the cell intrinsic effects of Ach in RG/oRG cells.

Developmental expression and gliogenesis. The expression of nAChR subunits encoded by CHRNA7 and CHRFA7A peaks in the germinal zone at GW23 (see Extended Figure 2C), a period of high gliogenic activity. Does modulating nAChR activity or expression in late progenitors also affect gliogenesis (reduction or delayed gliogenesis)?

Data visualization. I recommend presenting all quantifications in figures as violin plots rather than dot clouds, as the latter can be difficult to interpret.

Figure 3 – Differential effects of ACh and nicotine. Treatment with ACh and nicotine does not always produce similar effects, likely because mAChRs are not sensitive to nicotine. It would be useful to specifically assess how endogenous ACh/choline influences cortical progenitor biology and to what extent some of its effects are mediated via the mAChR G-protein-coupled receptors.

Figure 3 – Impact on neurogenesis. Activation of nAChRs/mAChRs likely shifts the balance between progenitor self-renewal and neuronal differentiation. However, neuronal reduction is assessed using NeuN, which may not be the earliest neuronal marker. Could it be that neurons are generated but experience delayed differentiation (which may also be the case for gliogenesis latter)? Additionally, can the authors directly test the impact of ACh/nicotine on progenitor cell cycle exit and the generation of newborn neurons?

As reported in the text and in figure 3. How does activation of nAChR receptors on organotypic slices from GW18-19 - when mostly upper layer neurons are generated - leads to drop in production (no cell death reported) of deep layer Ctip2+ neurons?

Figure 4. The discrepancy about the reduction of deep and upper layer neurons after pharmacological activation of nAChRs

between human cortical slice and cerebral organoids is questionable. Should the organoid model protocol be adapted to better match the studied human cortex stages? In addition, Satb2 staining should be included in the corresponding figure.

Mechanism of YAP1 regulation. How does nAChR activation lead to changes in YAP1 subcellular distribution and gene expression?

Minor comments

Figure 1a: Indicate the anterior-posterior (A-P) and lateral-medial (L-M) orientation of the forebrain section in the panels. The same applies to Extended Data Figure 1. The overview picture at GW15 is not convincing enough to show the colocalization of AChE and Hopx (no zoom-in picture was provided to confirm).

Figure 1b: Label the marginal zone (MZ) in the panel, as it is referenced in the main text.

Page 4, line 127: The reference to Figure 2b is incorrect in the statement "...with reduced levels at GW23 (Fig. 2b)." Please revise accordingly.

Figure 2b: The insets appear overly magnified (scale bar: 50 μ m) relative to the corresponding squared area in the larger field image. This also contrasts with Figure 2a and its magnified insets. In addition, it looks like little of expression *CHRNA7* and *CHRFAM7A* are not only in Sox2+ progenitors. What are the rest of the cells expressing it?

Figure 5b: Why are there so few Sox2+ cells, with almost none appearing uninfected? Additionally, where are the insets taken along the apico-basal axis of the cortical wall?

Figure 7d: Labelling in DCH1 staining: a box in the oSVZ overview is missing. As well as scale bar on the overview is missing. For YAP1 staining: it is a little confusing, better to place the oSVZ overview and zoom-in next to each other. Scale bars are missing.

Version 1:

Reviewer comments:

Reviewer #1

(Remarks to the Author)

The authors have responded to my queries and presented new supporting evidence. The manuscript has now the quality for publication in Nature Communications.

Reviewer #2

(Remarks to the Author)

I appreciated the authors' efforts in thoroughly addressing the comments. The manuscript has been significantly improved, and I fully support its publication.

Reviewer #3

(Remarks to the Author)

The authors provide a rebuttal with most answers to my questions. This also includes an update of existing figures and text modifications. As a last minor request, the authors could make a stronger point regarding the physiology of the receptors in cortical progenitors by relying on previous study from the lab and referring to the Mayer paper in the first paragraph of the results.

Reviewer #4

(Remarks to the Author)

REVIEWER COMMENTS (Black font)
REBUTTAL COMMENTS (Purple font)

Reviewer #1 (Remarks to the Author):

This paper investigates the effects prenatal nicotine exposure on human neuronal development. The authors used primary organotypic slices and organoid cortical cultures to investigate this question. The data obtained by the authors point at the participation of CHRNA7 and CHRFA7A nicotinic subunits in this process. These receptors use YAP1 for downstream signaling. The main results are that nAChR activation disbalances the ratio between neural progenitors and neurons by decreasing the neuronal population in the developing cortex. In line, loss-of-function experiments using sh-RNAs induced inverse effects compared to those promoted by CHRNA7 and CHRFA7A activation. Knock-down of CHRNA7 and CHRFA7A results in upregulation of YAP1 expression. The manuscript uses an excessive number of methods, including knock-down and pharmacological treatments, RNA scope, bulk and single cell RNAseq and bioinformatics analysis. Statistical analysis is adequate. The work is significant and meets the expected standards for publication.

Some points, which might be useful:

1.The significance of the work is clear for human health, as premature nicotine exposure affects cortical development. However, how the chosen nicotine dose reflects conditions of nicotine concentration in the premature brain?

We appreciate the reviewer's comment. The nicotine concentration of 500µM was chosen based on commonly reported concentrations in the literature, which have been shown to elicit maximal neuronal activity^{1,2}. In active pregnant smokers, plasma nicotine levels are typically $\geq 11.5\text{ng/ml}$ ^{3,4}. Additionally, nicotine levels in the developing fetus have been found to be 15 times higher than those in maternal plasma⁵.

We repeated our nicotine exposure experiments by exposing primary cortical cultures with low (10µM) concentrations of nicotine for 5 days, followed by single-cell sequencing using the Split-seq method (Rebuttal Figure 1a, as detailed in the Methods section of the manuscript). We collected approximately 20,000 cells from two biological replicates and their corresponding untreated controls. UMAP clustering revealed distinct, well-known cell clusters (Rebuttal Figure 1b), including radial glia, oRG, and various neuronal subtypes. All cell clusters expressed established marker genes (Rebuttal Figure 1c).

Following the lower-level nicotine exposure, differential expression analysis identified several key genes, such as effectors from the Semaphorin-Plexin pathway and the Hippo ligand *DCHS1*. These changes in radial glia (RG) and Layer 5/6 excitatory neurons (EN) align with our previous findings (Rebuttal Figure 1d-g). Gene ontology analysis highlighted processes like Semaphorin-Plexin signaling, cell adhesion, and cytoskeletal organization in RG, while Layer 5/6 excitatory neurons showed alterations in post-synaptic density, microtubule organization, and synapse development. Both high and low nicotine concentrations (with the latter reflecting reported exposure levels) resulted in consistent effects on RG and EN, including changes in the Hippo signaling effector *DCHS1*, further supporting the mechanistic conclusion of our findings.

Rebuttal Figure 1 and Extended Data Fig. 8. Differential Gene expression with levels of Nicotine approximating fetal exposure

Rebuttal Figure 1. Differential gene expression with levels of Nicotine approximating fetal exposure.

a. Workflow showing Nicotine exposure in primary dissociated cortical cell cultures. Cells were fixed on day 5 following the Parse Biosciences Fixation protocol and processed for Split-sequencing, a combinatorial barcoding approach where different barcodes are added in each step. Gene expression libraries were prepared and sequenced using Illumina NovaSeq2 Flow cell. b. UMAP clustering of single-cell data identifies expected cell types. c. Preliminary sc-analyses reveals anticipated patterns of marker gene expression across clusters. SOX2, PAX6, CRYAB, HOPX, LIFR, YAP1, VIM are expressed in radial glia and oRG clusters, while RBFOX3, BCL11B and SATB2 are expressed by neuronal clusters, LHX6, NR2F2, are expressed in Interneurons, S100B, AQP4 and OLIG2 in astrocytes and oligodendrocytes. d. Genes downregulated upon NIC exposure in RG cells. e. Gene Ontology (GO) for dividing RG identifies significant pathways and processes upregulated post NIC exposure, including Semaphorin-plexin and cell adhesion signaling. f. Genes downregulated upon NIC exposure in Layer 4/5 Excitatory neurons. One of the top downregulated genes is *DCHS1*, a canonical ligand of Hippo signaling. g. Gene Ontology (GO) for Layer 4/5 Excitatory neurons identifies significant pathways and processes upregulated post NIC exposure, including postsynaptic density organization, microtubule organization, and synapse development. All DEGs are listed in Extended datasheet 3.

2. Nicotine is a selective nAChR agonist, while acetylcholine is also a mAChR agonist. Further, nicotine is more stable than acetylcholine in differentiation assays. I do not understand why acetylcholine was used instead of choline, which would activate alpha7 receptors. Further, a functional calcium imaging assay, confirming the functionality of these receptors and possible differences between cell responses would be helpful. Anyway, which nicotine concentration was used for the studies.

We specifically focused on nAChRs, as the central goal of our study was to model prenatal nicotine exposure in humans. Acetylcholine (ACh) was included as an endogenous ligand that activates nAChRs. Although ACh can also activate muscarinic receptors (mAChRs), these receptors are not highly expressed at this developmental stage (Rebuttal Figure 2)⁶, making their activation less relevant to our study.

[REDACTED]

Rebuttal Figure 2. Muscarinic receptors are expressed at low levels across cortical lineage at second trimester. Feature plots for key muscarinic receptors including CHRM1, CHRM2, CHRM3, CHRM4 and CHRM5 show minimal RNA expression across diverse cell types⁶.

We recognize that choline is an endogenous, specific activator of the $\alpha 7$ nAChR. However, in our study, we chose PNU as an alternative selective $\alpha 7$ nAChR agonist to specifically investigate the role of this receptor subtype⁷. Previous studies have examined the functionality of nAChRs in the cortex^{8,9}, including work from our lab that used calcium imaging to explore cortical responses to acetylcholine (ACh) in the germinal zone (which is enriched in radial glia (RG) and outer radial glia (oRG) and the

cortical plate. This study revealed lower, but still significant, activity in these regions (Figure from Mayer *et al*, 2019, C-D)⁸. For the current manuscript, we opted to focus on cell fate outcomes rather than acute receptor activation⁸. To assess functional changes in cell fate, we employed immunostaining and single-cell RNA sequencing (scRNA-seq) as our primary tools.

Additionally, we used a 100µM ACh concentration and 10µM and 500µM nicotine concentrations¹, based on previous research demonstrating significant outward currents at these levels. By using these concentrations, we aimed to better understand how different nAChR activations influence the cellular processes involved in fate determination, offering insights into the broader implications of receptor activation in cortical development.

Figure from Mayer *et al*, 2019⁸

[REDACTED]

3. Astrocytes, the most abundant glial cell type in the brain, are underrepresented in traditional cortical organoid models due to the delayed onset of cortical gliogenesis. In view of that, astrocytes in cortical development was not assessed in this paper.

We agree with the reviewer that cortical gliogenesis was not the primary focus of our study. Our preliminary analyses of snRNA-seq data revealed lower expression of nAChRs in glial cells, including Astrocytes and Oligodendrocytes (Rebuttal Figure 3a-c). We also investigated the effects of nicotine and acetylcholine on both neurogenesis and gliogenesis in primary cortical tissue. While we observed changes in neurons, we did not find any significant effects on astrocytes or oligodendrocytes following nicotine and acetylcholine treatment (Rebuttal Figure 3a-c). Based on these initial results, we decided to concentrate our efforts on the neurogenic lineage in both primary cortical tissue and cortical organoid models.

To further investigate the impact of acetylcholine and nicotine on gliogenesis, we have included images and quantifications of astrocytes (labeled with S100B) and oligodendrocytes (labeled with OLIG2) in Extended Data Figures 4c and 4d. Our findings show no significant changes in S100B and OLIG2 expression during the 5-day chase after agonist treatment. However, we acknowledge that previous studies have observed changes in astrocytes and oligodendrocytes following nicotine exposure^{10,11}. Additionally, we assessed the effects on astrocytes and oligodendrocytes in our nAChR knockdown

experiments and found a significant reduction in S100B- and OLIG2-positive cells upon knockdown of both CHRNA7 and CHRFA7A (Rebuttal Figure 3'b-d, now incorporated in Fig.5c,f, and Extended Data Figures 4c,d).

Rebuttal Figure 3. Impact of AChR activation on glial cells

a Glial cells express low levels of AChRs

d

e

Rebuttal Figure 3. Impact of nAChR activation on astrocytes and oligodendrocytes.

a. UMAP clustering of single-cell data from Wang and Wang *et al.*, 2024¹², identifies expected cell types. b,c. Feature plots showing CHRNA7 and CHRFA7A expression across cortical development. d. Representative images of TUJ1-positive cells following co-immunostained with S100B (astrocyte marker), and OLIG2 (oligodendrocyte marker) with DAPI (nuclei). e. Quantification of S100B and OLIG2, reveals no significant changes following AChR activation, (n=3). Extended datasheet 1. *p = 0.05, **p = 0.01, ns = not significant.

Rebuttal Figure 3'. Impact of AChR manipulation on astrocytes and oligodendrocytes

Rebuttal Figure 3'. Impact of nAChR loss on astrocytes and oligodendrocytes.
a. Overview image of GW17 cortical tissue showing an example of lentiviral infection with shRNA targeting CHRNA7, co-immunostained for RFP (indicating shCHRNA7), SOX2 (marking radial glia and oRG), and DAPI (for nuclei), scale bar = 150 μ m. The boxes reflect the oSVZ and CP regions where the imaging and quantifications were made. **b, c.** Representative images of RFP-positive cells following nAChR knockdown, co-immunostained with S100B (astrocyte marker), OLIG2 (oligodendrocyte marker), and NEUROD1 (intermediate progenitor and immature neurons) (white arrows). **d.** Quantification of S100B, OLIG2, and NEUROD1 reveals significant changes following AChR knockdown (n=3). Extended datasheet 1. *p = 0.05, **p = 0.01, ns = not significant. LV- lateral ventricle, VZ- ventricular zone, ISVZ- inner subventricular zone, OSVZ- Outer subventricular zone, CP- Cortical plate.

4. Authors state that they use iPSC-derived cortical organoids. In the Methods' section two iPS cell lines (one male and one female) and a human embryonic stem cell line. However, i.e. Figure legend 2 does not state which cell line was used. Were any differences obtained between male and female iPSC-derived progenitors?

All the experiments using iPSC-derived cortical organoids were performed using 3 cell lines. We have updated the Figure legends to reflect the cell lines used. With only three cell lines, it is not possible to perform statistically meaningful comparisons between male and female samples. However, we included both male and female cell lines to represent both genders in our study. This would be an interesting direction for a future study but would require a larger number of biological samples to reach statistical significance.

5. Authors performed multiplexing (simultaneous staining) for Ki-67 expression with cellular phenotyping and CHRFAM7A expression? This is not stated in the Methods' section. This can be done by imaging cytometry and much more easily using state-of-the-art flow cytometers.

We have performed co-immunostaining of KI67 and CHRFAM7A (Rebuttal Figure 4), showing representative double positive cells for KI67 and CHRFAM7A (marked with white arrowheads). We have updated the Methods section with this information in the manuscript and incorporated these results in Fig. 2.

Rebuttal Figure 4

CHRFAM7A protein is expressed in KI67 positive cells in neurogenic cortical organoids

Week 7-8 Cortical Organoids

Rebuttal Fig. 4. CHRFAM7A protein is expressed in KI67 positive proliferating cells in neurogenic cortical organoids. Representative images of week 7-8 cortical organoids show expression of CHRFAM7A in KI67 positive cells (white arrowheads). Scale bar, 50um, and 25um for the magnified inset.

6. The Western blots, especially for CHRNB1 and CHRNB2, in Extended Data Fig. 2 are not acceptable. Which negative control was used to probe for the selectivity of the antibody? Why are the staining intensities for the GAPDH control different, and why GAPDH shows a double band in one occasion (d) and a single band in (e) and (f).

We acknowledge the reviewer's concern regarding the Western blots; however, GAPDH is known to exist in multiple isoforms in humans, as shown in Rebuttal Fig. 5a. Furthermore, we used a commercially available antibody for GAPDH, and several prior studies have reported the presence of multiple bands for GAPDH in Western blot analyses of human sample lysates^{13,14}. Other potential explanations for the observed bands could include post-translational modifications or protein degradation. The primary band observed matches the expected molecular weight, though we also noted faint additional bands in some samples (Extended Data Fig. 2e and f). These additional bands may represent protein processing variants, but further investigation would be needed to determine their precise nature.

It is also worth noting that we observe a single band in whole cortical lysates, but a more prominent smaller band in enriched germinal zone and cortical organoid lysates. Since we did not use Western blotting for quantification but rather to confirm the presence or absence of the protein, the varying band intensities were not the primary focus of our analysis.

For the nAChRs, we have also used commercially available, manufacturer-validated antibodies (details provided below).

CHRNA7	Rabbit	Alomone labs	ANC-007
CHRFAM7A	Rabbit	Aviva system biology	OAAN02465
CHRNA1	Rabbit	Proteintech	11553-1-AP
CHRNA2	Rabbit	Abcam	ab41174

NACHRs also exist as multiple isoforms (Rebuttal Figure 5b-e). The additional bands observed for CHRNA1 and CHRNA2 may be due to isoform diversity, post-translational modifications, or partial protein degradation. Previous publications have also observed additional bands in CHRNA1, CHRNA2:

- 1) Fig. 6B, shows additional bands in CHRNA1¹⁵,
- 2) Fig. 4D, E show additional bands in CHRNA2¹⁶,
- 3) Fig. 2C, shows additional bands in CHRNA7¹⁷) and
- 4) Knockout validated antibodies for CHRNA7/CHRFAM7A¹⁸.

Rebuttal Figure 5.
isoform diversity in GAPDH and nAChRs

Rebuttal Figure 5. Isoform diversity in GAPDH and nAChRs. Known isoforms for a. GAPDH, b. CHRNA7, c. CHRFAM7A, d. CHRNA1 and e. CHRNA2 (as shown using UCSC genome browser) reveal notable transcript diversity.

7. The authors need to comment on the limits of their work, compared to *in vivo* conditions.

As a substitute for *in vivo* human studies, we used human *ex vivo* organotypic slices and organoids to model prenatal nicotine exposure. While these systems offer valuable insights, they do have limitations. For example, the exposure duration is restricted by the long-term viability of the organotypic slices, and certain brain regions, that may contribute to clinical phenotypes, are not represented in our cultures. We have addressed these limitations in our discussion.

8. There are some inaccuracies in the writing style, please, check i.e. “Activation of nAChRs increases progenitors and decreases neurons in developing human cortex”. “NIC is known to primarily act on nAChRs while ACh also acts on muscarinic acetylcholine receptors.” There is also an excess of abbreviations. Why nicotine needs to be abbreviated as “NIC”?

We thank the reviewer for pointing this out and we have updated the text accordingly.

Reviewer #2 (Remarks to the Author):

In this work, Mukhtar et al., studied the role of $\alpha 7$ nicotinic acetylcholine receptors in regulating cell fate in the developing human cortex. They focused on two nAChR subunits, CHRNA7 and CHRFA7A, expressed in both progenitors and neurons of fetal cortex. They modeled prenatal exposure to nicotine (NIC) by activating these receptors pharmacologically and studied their role during human cortical development by protein quantification, bulk and scRNA-seq in both fetal cortices' slices and cortical cell cultures. Their findings were corroborated by nAChR knockdown experiments. The study demonstrated that activation of these receptors results in an increase of progenitor cells and radial glia and a decrease of both deep and upper-layer neurons. Although previous studies on prenatal NIC have been conducted in rodents and non-human primates, this study is unique in its exploration of these mechanisms in human fetal brain specimens. I believe it holds significant importance for the neurodevelopmental field, particularly because it focuses on two nAChRs previously associated with neurological disorders like SCZ and BD. Overall, the study is of great interest, proposing a novel signaling interaction between Hippo pathway and cholinergic signaling, though improvement on a few points would make it an outstanding study.

Major comments

The authors investigated CHRFA7A gene and protein expression in both *ex vivo* and organoid slices but did not do so for CHRNA7 (Figure 2). Which antibody was used—was it the same one used for the western blot assay? Why was immunofluorescence for CHRNA7 not included? Text.

Immunostaining for CHRNA7 was carried out on cortical tissue sections from GW14, 16, 18, 20, and 23, spanning neurodevelopment (Extended Figure 3). We observed CHRNA7 expression in SOX2-positive cells in the Ventricular/Subventricular zone (VZ/ISVZ) and Outer Subventricular zone (OSVZ), as indicated by arrowheads. Additionally, CHRNA7 protein expression was detected in the Cortical plate (CP).

We used a commercially available rabbit polyclonal antibody for CHRNA7 (Cat No. ANC007, Alomone Labs) and noted some background signal, examples of which are shown at GW18 and GW23 (marked by arrowheads in Fig 2b, b'). To confirm the specificity of the staining, we compared the staining using

RNAscope for CHRNA7, which replicated these results. The validated immunostaining data is now included in the manuscript in Extended Figure 3.

Rebuttal Figure 6.
NACHR subunit CHRNA7 is expressed in Germinal zones

Rebuttal Figure 6.
NACHR subunit CHRNA7 is expressed in Germinal zones across human cortical development.

Representative images from VZ to CP show co-immunostaining for CHRNA7 (nAChR), SOX2 (radial glia, outer radial glia), and DAPI (nuclei) in cortical sections from GW14, GW16, GW18, GW20, and GW23, scale bar = 500um. Magnified images highlight the expression of CHRNA7 protein (arrowheads) in the VZ, oSVZ, and CP throughout development, scale bar = 50um. VZ-Ventricular zone, iSVZ-Inner subventricular zone, OSVZ-Outer subventricular zone, CP-Cortical plate.

Regarding protein analysis by western blot, could the authors comment on the presence of additional bands in Ex. Fig. 2c-f? Are these known isoforms? In Ex. Fig. 2c-d, additional bands of both higher and lower molecular weights are observed in some samples. Were these antibodies previously validated?

We acknowledge the reviewer's concern regarding the Western blots; for the nAChRs, we have used commercially available, manufacturer-validated antibodies (details provided below):

CHRNA7	Rabbit	Alomone labs	ANC-007
CHRFAM7A	Rabbit	Aviva system biology	OAAAN02465
CHRNA7	Rabbit	Proteintech	11553-1-AP
CHRNA7	Rabbit	Abcam	ab41174

nAChRs exist as multiple isoforms (Rebuttal Figure 5b-e, also explained above, in Reviewer 1, Q6). The additional bands observed for CHRNA7 and CHRNA8 may be due to isoform diversity, post-translational modifications, or partial protein degradation. Previous publications have also observed additional bands in CHRNA7, CHRNA8:

- 1) Fig. 6B, shows additional bands in CHRNA7¹⁵,
- 2) Fig. 4D, E show additional bands in CHRNA8¹⁶,
- 3) Fig. 2C, shows additional bands in CHRNA7¹⁷) and
- 4) Knockout validated antibodies for CHRNA7/CHRFAM7A¹⁸.

Rebuttal Figure 5.
isoform diversity in GAPDH and nAChRs

Rebuttal Figure 5. Isoform diversity in GAPDH and nAChRs. Known isoforms for a. GAPDH, b. CHRNA7, c. CHRFAM7A, d. CHRNA7 and e. CHRNA8 (as shown using UCSC genome browser) reveal notable transcript diversity.

The molecular weight of CHRFBAM7A is unclear, as it appears to be 50 kDa in panel c but 55 kDa in panel d. Please verify and correct this discrepancy. Also, in Ex. Figure 2f, a completely different molecular weight (100 kDa) is reported for CHRFBAM7A in the organoid lysate. Why does this discrepancy exist? Please check and clarify.

In the manuscript, page 4, the authors claim higher expression of CHRFBAM7A in the germinal zone. Quantification of the protein bands from the germinal lysates would strengthen this claim.

We thank the reviewer for this comment. It was an oversight on our part, and we have now updated the information. Also, nAChRs exist as multiple isoforms (Rebuttal Figure 5b-e, also explained above, also in Reviewer 1, Q6). The additional bands observed may be due to isoform diversity, post-translational modifications, or partial protein degradation. Previous publications have also observed additional bands in CHRNA1, CHRNA2:

- 1) Fig. 6B, shows additional bands in CHRNA1¹⁵,
- 2) Fig. 4D, E show additional bands in CHRNA2¹⁶,
- 3) Fig. 2C, shows additional bands in CHRNA7¹⁷) and
- 4) Knockout validated antibodies for CHRNA7/CHRFBAM7A¹⁸.

We observed high expression of CHRFBAM7A along the lateral ventricle using BaseScope and immunostaining, which may suggest a non-canonical role for these nAChRs in radial glia (RG). The reference on page 4 pertains to BaseScope expression along the lateral ventricle across various temporal samples. We opted not to use Western blotting for quantification, as the germinal zone contains not only RG, IPCs, and oRG, but also immature, migratory excitatory neurons and interneurons. Consequently, protein expression measured in these samples would not be exclusive to RG, IPCs, and oRG but would also include contributions from migratory excitatory neurons and interneurons. Instead, we used Western blotting to confirm the presence or absence of the protein, with varying band intensities not being the primary focus of our analysis.

The pharmacological functional studies of the cholinergic system were elegantly designed, with appropriate controls and a TUNEL assay to exclude treatment-induced cell death or apoptosis. However, few points require clarification:

-What criteria or references were used to determine the concentrations of the small molecules in the study? REF.

We appreciate the reviewer's comment. We used a 100µM ACh concentration and 10µM and 500µM nicotine concentrations¹, based on previous research demonstrating significant outward currents at these levels. By using these concentrations, we aimed to better understand how different nAChR activations influence the cellular processes involved in fate determination, offering insights into the broader implications of receptor activation in cortical development. For more details, please see Reviewer 1, Q1, and Q2.

-Can the authors comment on the EOMES results? While both molecules increased progenitor proliferation (SOX2+) and downregulated neuronal markers (NEUN, CTIP2, and SATB2), why did NIC uniquely favor the EOMES fate?

Both nicotine and acetylcholine increased progenitor proliferation (SOX2+) and downregulated neuronal markers (NEUN, CTIP2, and SATB2). However, NIC uniquely favored the EOMES fate, which may be attributed to differences in receptor activation and downstream signaling effects.

Nicotine is a more potent agonist for nAChRs than ACh, potentially leading to stronger activation of nAChR-mediated pathways¹⁹. One possible explanation for the differential increase in EOMES-positive cells is that Nicotine may promote a more robust translocation of YAP1 into the nucleus, potentially activating YAP1-associated genes (Fig. 7b and 7f). Given the impact of Hippo signaling in our study, this mechanism could explain Nicotine's selective influence on EOMES-positive intermediate progenitors.

Supporting this hypothesis, overexpression of YAP1 in murine embryonic neural stem cells has been shown to increase PAX6-positive cells in the VZ/SVZ or SVZ alone, as observed in Supplementary Fig. 4D²⁰. This suggests that Nicotine-induced YAP1 activation may enhance progenitor expansion and direct differentiation toward the EOMES lineage.

-NIC exposure has previously been associated with GABAergic alterations. Besides DLX5 quantification (Ex. Fig. 3), considering that the authors focused only on cortical samples, this model lacks the major interneuron source, the basal ganglia. Given this limitation, the authors should be cautious in drawing conclusions regarding NIC's effects on this cell type, which was not fully investigated.

We agree with the reviewer's observation. Our current study focused exclusively on the dorsal cortex, and we did not investigate the effects of Nicotine exposure on the basal ganglia, which is the primary source of cortical interneurons. In light of this limitation, we are now more cautious in interpreting Nicotine's effects on GABAergic populations. We have updated the revised manuscript to acknowledge this limitation and provide a clearer explanation of the scope of our findings.

-The data involving the antagonist BTX in combination with NIC and ACh are less convincing and suggest that ACh and NIC actions are not exclusively mediated by the nAChRs studied. The authors themselves mention this on page 7. This limitation should be explicitly addressed in the discussion.

We agree with the reviewer's observation. We have updated the revised manuscript to reflect this and added a section to address this in the discussion.

-Do NIC and acetylcholine treatments alter the expression of different nAChRs in cortices and/or organoids?

We did not observe any significant changes in the RNA expression of CHRNA7 and CHRFA7A following treatment with nicotine and acetylcholine.

Rebuttal Figure 7. Acetylcholine and Nicotine do not affect CHRNA7 and CHRFA7A RNA expression. Bulk-RNA sequencing conducted on RNA extracted from dissociated cortical samples treated with Acetylcholine and Nicotine revealed no significant changes in nAChR expression following treatment.

The protein distribution of AChE in cortical tissue (Fig. 1) was well studied. Do cortical organoids express AChE?

We analyzed unpublished organoid datasets from our lab and observed AChE expression in some intermediate progenitor cells (IPCs) and excitatory neurons (ENs) (see in Rebuttal Fig. 8a-c). However, when visualized in a dot plot, AChE expression appears relatively low, which is consistent with the fact that *in vivo*, AChE-expressing fibers mainly originate from the nucleus basalis of Meynert, located outside the cortex²¹. This supports previous findings that cortical cholinergic innervation primarily comes from subcortical sources rather than from intrinsic cortical expression of AChE.

Rebuttal Figure 8. AChE expression in iPSC-derived cortical organoids

Rebuttal Figure 8. Acetylcholinesterase (AChE) expression in iPSC-derived cortical organoids. a. UMAP clustering reveals the known cellular diversity in iPSC-derived cortical organoids (data from Mukhtar and Wang et al., 2025, in preparation). b. Dot plots confirm the expression of well-established cell type-specific markers. c. Feature plot showing AChE expression in IPCs and excitatory neurons (ENs). EN - Excitatory neurons, RG - Radial glia, IPCs - Intermediate progenitor cells, oRG - Outer radial glia.

Did the authors investigate the long-term effects of this initial increase in progenitors? In cortical

organoids. Does this initial insult eventually lead to an increase in neurons, or is the aberrant overproduction of progenitors compensated for over time?

How was the 4-days exposure paradigm chosen?

We selected a short-term, 4-day exposure paradigm to model acute exposures, consistent with the *in vivo* timeline for primary tissue. We can only maintain organotypic slice cultures in healthy conditions for up to 7 days, and since the slices require a day to stabilize, previous experiments in our lab are typically restricted to 5 days to ensure optimal tissue quality for downstream analyses. During this period, we observed changes in cell fate and alterations in Hippo signaling, suggesting that significant molecular and cellular responses occur within this timeframe.

We acknowledge that investigating longer-term effects in organoids would be valuable. Future studies could extend the exposure duration to assess whether the initial increase in progenitors leads to an excess of neurons or if it is compensated for over time. While our primary focus was on *ex vivo* tissue, incorporating long-term organoid studies could provide additional insights into the prolonged impact of nicotine exposure on cortical development.

RNA bulk and scRNA sequencing data are interesting, highlighting several targets of the nAChR, but the text and the supporting figures need some editing. To mention a few examples, the heatmap Ext.Fig 5c gene labels are too small; the gene CACNA1A mentioned by the author as upregulated by shCHRNA7 is hard to be found in any Table/Data and if in the heatmap it is not visible. Only found in the qPCR validation tab and Ext.Fig.5g. Please provide better panels in the figure that can show these findings.

We appreciate the reviewer's comment and apologize for the oversight. We have updated the figures and text accordingly. We do observe an upregulation of CACNA1A following CHRNA7 knockdown, which has also been validated using RT-qPCR. Additionally, we have improved the heatmap and included all the candidate genes to better reflect the observed changes.

Bulk RNA sequencing	Gene Name	logFC	P Value
GW21_shCHRNA7_vs_shcontr_edgeR_output.txt:ENSG00000141837	CACNA1A	3.219699	0.002812
GW22_shCHRNA7_vs_shcontr_edgeR_output.txt:ENSG00000141837	CACNA1A	3.873257	0.005479

The investigation of molecular changes after NIC exposure by scRNA-seq, per cell types corroborate the previous finding of its role in cortical development followed by the discovery of inter-talk between nAChRs and YAP pathway make the paper very important, highlighting the necessity of further study for developmental disorders of these receptors, also in other cell types.

Minor comments

Authors might want to consider including a panel with the experimental design, tissue and organoids developmental stages analyzed in Figure 1.

We thank the reviewer for this comment. All of our figures have been updated with revised illustrations for the experimental design.

In Extended Figure 2d, the authors could add a small legend clarifying the distinction between "germinal zone lysate" and "whole cortices lysate" shown in panel c.

We have updated the Extended Figure 2d with the recommended changes.

In Figure 4b, the brightness of CTIP2 panels should be increased to improve the visualization of CTIP2+ cells or, alternative, the channel color could be changed to gray instead of yellow. The same applies to Extended Figure 4b.

The new Figure 4b has updated images of CTIP2 positive cells.

In the Loss-of Function nAChRs experiments using shRNAs (page 7), the authors should clarify at the beginning of the paragraph that the assay was first performed on dissociated cortical culture and then on organotypic slices. The EOMES+ results (line 273) are described as consistent with previous observations. However, I disagree, as NIC treatment increases the EOMES population, while PNU shows no change. Please verify this claim.

We have updated the text and made the necessary changes.

In the discussion (page 14), authors claim to have identified twice as many exclusive DEGs for CHRFAM7A knockdown. Please curate the Tables and reference a corresponding Figure (Fig.6g-h?).

We thank the reviewer for this comment. We have added the plot to Extended Datasheet 3.

A legend tab is highly recommended for all Tables.

The updated manuscript now contains detailed legends for all Extended Datasheets on their first tab.

The Material and Methods section is well written, though under Small molecule treatments the concentration of the first molecule listed (NIC) is missing.

We have now added the necessary information.

Reviewer #3 (Remarks to the Author):

Prenatal nicotine exposure has been associated with a reduction in fetal cortical gray matter volume. To elucidate the underlying mechanisms, researchers investigated the role of two nicotinic acetylcholine receptor (nAChR) subunits, CHRNA7 and the human-specific CHRFAM7A, in human cortical progenitors and neurons. Their findings identified a novel function of nicotinic receptor signaling in regulating the proliferation, self-renewal, and neurogenesis of radial glial (RG) and outer radial glial (oRG) cells. More specifically, pharmacological activation of nAChRs, either through broad-spectrum or specific agonists, resulted in an increased number of RG cells while concomitantly reducing the populations of deep-layer and upper-layer neurons. Conversely, knockdown of nAChRs led to a reduction in RG cells and a corresponding increase in neuronal populations, though the precise neuronal subtypes affected remain to be determined. These findings indicate that CHRNA7 and CHRFAM7A play a crucial role in mediating fate transitions in response to prenatal nicotine exposure. scRNAseq analysis further revealed that nicotine exposure downregulates key genes essential for excitatory neuron function. Notably, many of these transcriptional alterations were uniquely regulated by either CHRNA7 or CHRFAM7A, suggesting an evolutionary divergence in their regulatory functions. Further investigation identified the Hippo signaling co-activator YAP1 as a downstream effector of nAChR activation, thereby establishing a mechanistic link between prenatal nicotine exposure and alterations in RG and oRG cell populations through YAP1-mediated signaling. These findings provide critical insights into how prenatal nicotine exposure disrupts cortical

development by influencing progenitor cell fate decisions and excitatory neuron gene expression. By distinguishing the distinct roles of CHRNA7 and CHRFA7A, the study highlights an evolutionary divergence in cholinergic regulation of neurodevelopment. Collectively, the dataset and conclusions presented are scientifically robust and contribute to a deeper understanding of the impact of nicotine on the developing brain.

Major comments

Electrophysiological analysis of nAChR function is missing. What are the membrane effects of activating the $\alpha 7$ homomeric nAChR and the human-specific variant formed by the CHRFA7A-encoded subunit? Do they induce depolarization in RG and oRG cells, thereby activating VGCCs and triggering calcium-dependent downstream molecular signaling? This would also be important to support the cell intrinsic effects of Ach in RG/oRG cells.

We acknowledge that electrophysiological analysis of nAChR function was not included in this study. Previous studies have examined the functionality of nAChRs in the cortex^{8,9}, including work from our lab that used calcium imaging to explore cortical responses to acetylcholine (ACh) in the germinal zone (which is enriched in radial glia (RG) and outer radial glia (oRG) and the cortical plate (Reviewer 1, Q2, Figure from Mayer *et al*, 2019, C-D)⁸. Ca²⁺ imaging was conducted in organotypic slice cultures of the germinal zone (GZ), which is rich in radial glia (RG) and outer radial glia (oRG), as well as the ventricular zone (VZ), inner subventricular zone (iSVZ), outer subventricular zone (oSVZ), subplate, and cortical plate (CP)⁸. Our results revealed that acetylcholine (ACh) triggers a lower but significant release of Ca²⁺ in these regions (Fig. 3E,F), suggesting that ACh may regulate intracellular calcium levels during cortical development. These findings prompted us to further investigate the role of nicotinic acetylcholine receptors (nAChRs) in neural progenitors⁸.

The $\alpha 7$ nAChR is known for its high calcium permeability, allowing direct Ca²⁺ influx upon activation^{22,23}. Additionally, $\alpha 7$ nAChR activation can indirectly elevate intracellular Ca²⁺ levels by triggering voltage-dependent calcium channels (VDCCs) through neuronal depolarization²⁴ and triggering Ca²⁺-induced Ca²⁺ release (CICR) from the endoplasmic reticulum²⁵. However, we did not conduct patch-clamp recordings in RG or oRG cells to directly assess these effects in progenitors.

Regarding the human-specific CHRFA7A-encoded subunit, prior studies have demonstrated that CHRFA7A modulates $\alpha 7$ nAChR function by reducing calcium permeability, increasing desensitization, and altering responses to allosteric modulators²⁶. This suggests that individuals with varying levels of CHRFA7A expression may experience different nAChR-mediated depolarization and downstream signaling effects.

Electrophysiological studies of $\alpha 7$ nAChR function have mainly focused on neuronal cultures, with limited knowledge regarding functional effects in RG/oRG cells. While our study primarily addressed cell fate changes rather than acute electrophysiological effects, future studies incorporating calcium imaging or patch-clamp recordings in RG/oRG cells would be valuable for further understanding the membrane effects of $\alpha 7$ nAChR activation and CHRFA7A modulation.

Developmental expression and gliogenesis. The expression of nAChR subunits encoded by CHRNA7 and CHRFA7A peaks in the germinal zone at GW23 (see Extended Figure 2C), a period of high gliogenic activity. Does modulating nAChR activity or expression in late progenitors also affect gliogenesis (reduction or delayed gliogenesis)?

We apologize for any confusion. Extended Data Fig. 2c shows the expression of CHRNA7 and CHRFA7A in whole cortical lysates, which also include neurons. Since the cortices at GW23 are larger than those at GW17, the observed higher protein levels of CHRNA7 and CHRFA7A could be attributed to the difference in cortical sample sizes. Also, we did not use Western blotting for quantification but rather to confirm the presence or absence of the protein, the varying band intensities were not the primary focus of our analysis.

We agree that AChRs may play a functional role in gliogenesis, we have further explored this by performing immunostaining with GFAP and OLIG2 following knockdown to assess the impact on astrocytes and oligodendrocytes (Rebuttal Figure 3'b-d, also explained in Reviewer 1, Q3). We observed a reduction in S100B+ astrocytes and a significant decrease in OLIG2+ oligodendrocytes. This data has been added to Fig. 5 in the revised manuscript.

Rebuttal Figure 3'.
Impact of AChR manipulation on astrocytes and oligodendrocytes

Rebuttal Figure 3'. Impact of nAChR loss on astrocytes and oligodendrocytes.
a. Overview image of GW17 cortical tissue showing an example of lentiviral infection with shRNA targeting CHRNA7, co-immunostained for RFP (indicating shCHRNA7), SOX2 (marking radial glia and oRG), and DAPI (for nuclei), scale bar = 150 μ m. The boxes reflect the oSVZ and CP regions where the imaging and quantifications were made. **b, c.** Representative images of RFP-positive cells following nAChR knockdown, co-immunostained with S100B (astrocyte marker), OLIG2 (oligodendrocyte marker), and NEUROD1 (intermediate progenitor and immature neurons) (white arrows). **d.** Quantification of S100B, OLIG2, and NEUROD1 reveals significant changes following AChR knockdown (n=3). Extended datasheet 1. *p = 0.05, **p = 0.01, ns = not significant. LV- lateral ventricle, VZ- ventricular zone, ISVZ- inner subventricular zone, OSVZ- Outer subventricular zone, CP- Cortical plate.

Data visualization. I recommend presenting all quantifications in figures as violin plots rather than dot clouds, as the latter can be difficult to interpret.

We have changed all our quantification plots to Violin plots and updated the figures.

Figure 3 – Differential effects of ACh and nicotine. Treatment with ACh and nicotine does not always produce similar effects, likely because mAChRs are not sensitive to nicotine. It would be useful to specifically assess how endogenous ACh/choline influences cortical progenitor biology and to what extent some of its effects are mediated via the mAChR G-protein-coupled receptors.

While we acknowledge that mAChRs (CHRM1, CHRM2, CHRM3, CHRM4, and CHRM5) may contribute to some functions, our single-cell data indicates low expression of mAChRs across cortical cells at the timepoints used in our study (see below), suggesting that their role in this context may be limited. Given this, we focused on nAChRs as the primary cholinergic receptors in our study.

[REDACTED]

Rebuttal Figure 2. Muscarinic receptors are expressed at low levels across cortical lineage at second trimester. Feature plots for key muscarinic receptors including CHRM1, CHRM2, CHRM3, CHRM4 and CHRM5 show minimal RNA expression across diverse cell types at this age⁶.

Figure 3 – Impact on neurogenesis. Activation of nAChRs/mAChRs likely shifts the balance between progenitor self-renewal and neuronal differentiation. However, neuronal reduction is assessed using NeuN, which may not be the earliest neuronal marker. Could it be that neurons are generated but experience delayed differentiation (which may also be the case for gliogenesis later)? Additionally, can the authors directly test the impact of ACh/nicotine on progenitor cell cycle exit and the generation of newborn neurons?

We have conducted additional experiments and included the results in Extended Data Fig. 4b and Extended Datasheet 1 in the revised manuscript. Co-immunostaining for KI67 and EdU reveals an increase in double-positive cells following Acetylcholine and Nicotine treatment, indicating extended cell cycle length. Additionally, immunostaining for Beta-tubulin III (TUJ1), an earlier neuronal marker than NEUN, also showed a significant reduction after Nicotine exposure. Statistical significance is indicated as * $p=0.05$, ** $p=0.01$, *** $p=0.001$, **** $p=0.0001$.

As reported in the text and in figure 3. How does activation of nAChR receptors on organotypic slices from GW18-19 - when mostly upper layer neurons are generated - leads to drop in production (no cell death reported) of deep layer Ctip2+ neurons?

While neurons in deep cortical layers are generated prior to those in upper layers, there is considerable overlap, and the limits have not been well-defined in human corticogenesis. In our experiments, activation of nAChR receptors in Week 5-6 organoids led to a significant reduction in CTIP2-positive cells (Extended Data Fig. 5b,c). Genes that define neuronal subtypes and cortical layers in development—such as *Tbr1* and *Ctip2* (Layers V and VI), and *Satb2* and *Cux2* (Layers IV and II/III)—demonstrate that the generation of CTIP2+ cells may continue, albeit at a lower rate, even during the late stages of neurogenesis in mice²⁷. Studies using EdU injections followed by co-labeling with CTIP2 or SATB2 immunostaining have shown that newborn CTIP2+ neurons can still be detected as late as E18 in mice. At these later stages, the proportion of newborn CTIP2+ neurons is relatively small, which is consistent with the transcriptional dynamics and the changes in CTIP2+ cell numbers observed in our organoid model²⁸.

Figure 4. The discrepancy about the reduction of deep and upper layer neurons after pharmacological activation of nAChRs between human cortical slice and cerebral organoids is questionable. Should

the organoid model protocol adapted to better match the studied human cortex stages? In addition, Satb2 staining should be included in the corresponding figure.

We have now included SATB2 immunostaining in the updated figure 4.

Corticogenesis occurs during prenatal stages that are challenging to study directly in humans across all stages of neurogenesis. Furthermore, while human primary tissue offers the most accurate representation of the molecular principles of corticogenesis, its viability and cytoarchitectural organization decline after a week of *ex vivo* culture, also dependent on the gestational age. Therefore, the analysis of primary tissue must be complemented with an appropriate *in vitro* model. Cortical organoids derived from induced pluripotent stem cells offer a valuable model that replicates key aspects of cortical development *in vitro*. Our previous research has successfully used this system to study neuroepithelia (NE) and radial glia (RG) during corticogenesis and to model neurodevelopmental diseases²⁹. Studies have shown that cortical organoids, generated through various protocols, recapitulate essential features of corticogenesis, including cell proliferation, differentiation, and migration, in a reproducible manner. However, single-cell and morphological evaluations have highlighted certain limitations of the system, such as metabolic stress, the absence of external signaling cues and cell types (e.g. vasculature and microglia), as well as differences in cell proportions and structural organization²⁹. Despite these challenges, cortical organoids offer greater flexibility for manipulation and longitudinal study than primary tissue, enabling experiments that would not be feasible with primary tissue alone. Given these considerations, for our current study, we have strategically employed both primary cortical tissue and cortical organoids, leveraging the unique strengths of each model to enhance the robustness and depth of our findings.

Mechanism of YAP1 regulation. How does nAChR activation lead to changes in YAP1 subcellular distribution and gene expression?

Hippo signaling is an evolutionarily conserved signaling cascade involved in organ size control and tissue homeostasis³⁰. Reduction in expression of Fat tumour suppressor homologue (Fat4) and Dachous (Dchs), the upstream receptor and ligand of the Hippo pathway in embryonic neural stem cells, results in increased proliferation and reduction of neuronal differentiation³⁰. The Hippo pathway involves a series of phosphorylation steps mediated by the kinases, Macrophage stimulating-1/2 (Mst1/2) and Large tumour suppressor-1/2 (Lats1/2), that phosphorylate Yes-associated protein/Transcriptional co-activator with PDZ-binding motif (Yap1/Taz) co-activators and promote their retention in the cytoplasm and subsequent degradation. If Hippo is off, Yap/Taz translocate to the nucleus and bind the TEA domain transcription (Tead) TFs and regulate gene expression. Yap/Taz have multiple binding partners in the nucleus for example, SMADs, Runx etc³⁰.

Our single-cell sequencing data revealed differential expression of upstream Hippo signaling receptors (FAT3, CRB2) and ligands (DCHS1) following Nicotine exposure and nAChR knockdown. Loss of FAT and DCHS1 has been shown to upregulate the co-activator YAP1²⁰, and we observe a similar increase in YAP1 after Nicotine exposure. Additionally, the phenotypic changes, including an increase in SOX2, observed are consistent with our findings²⁰. In esophageal squamous cell cancer, Nicotine is known to upregulate YAP1 and increase its nuclear retention^{31,32}.

Minor comments

Figure 1a: Indicate the anterior-posterior (A-P) and lateral-medial (L-M) orientation of the forebrain section in the panels. The same applies to Extended Data Figure 1. The overview picture at GW15 is not convincing enough to show the colocalization of AChE and Hopx (no zoom-in picture was provided to confirm).

The figures are updated in the revised manuscript.

Figure 1b: Label the marginal zone (MZ) in the panel, as it is referenced in the main text.
The changes are incorporated in the revised manuscript.

Page 4, line 127: The reference to Figure 2b is incorrect in the statement "...with reduced levels at GW23 (Fig. 2b)." Please revise accordingly.
We apologize for this oversight; this has been updated in the revised manuscript.

Figure 2b: The insets appear overly magnified (scale bar: 50 μ m) relative to the corresponding squared area in the larger field image. This also contrasts with Figure 2a and its magnified insets. In addition, it looks like little of expression CHRNA7 and CHRFA7A are not only in Sox2+ progenitors. What are the rest of the cells expressing it?
We have updated the figures based on the reviewer's suggestions. The other cells may be migrating neurons, which are known to express low levels of nAChRs, as also observed in Extended Fig. 2a.

Figure 5b: Why are there so few Sox2+ cells, with almost none appearing uninfected? Additionally, where are the insets taken along the apico-basal axis of the cortical wall?
We have primarily focused on the oSVZ for our imaging and quantification. We have now included an example tile scan in Fig. 5b, showing the full extent of the tissue sections, highlighting the areas that were imaged.

Figure 7d: Labelling in DCHS1 staining: a box in the oSVZ overview is missing. As well as scale bar on the overview is missing. For YAP1 staining it is a little confusing, better to place the oSVZ overview and zoom-in next to each other. Scale bars are missing.
We thank the reviewer for this comment, we have now updated this in the revised manuscript.

References

- 1 Weir, K., Dupre, C., van Giesen, L., Lee, A. S. & Bellono, N. W. A molecular filter for the cnidarian stinging response. *Elife* **9**, doi:10.7554/eLife.57578 (2020).
- 2 Takahashi, T. *et al.* Component of nicotine-induced intracellular calcium elevation mediated through alpha3- and alpha5-containing nicotinic acetylcholine receptors are regulated by cyclic AMP in SH-SY 5Y cells. *PLoS One* **15**, e0242349, doi:10.1371/journal.pone.0242349 (2020).
- 3 Minatoya, M. *et al.* Prenatal tobacco exposure and ADHD symptoms at pre-school age: the Hokkaido Study on Environment and Children's Health. *Environ Health Prev Med* **24**, 74, doi:10.1186/s12199-019-0834-4 (2019).
- 4 Kvalvik, L. G. *et al.* Self-reported smoking status and plasma cotinine concentrations among pregnant women in the Norwegian Mother and Child Cohort Study. *Pediatr Res* **72**, 101-107, doi:10.1038/pr.2012.36 (2012).
- 5 Lambers, D. S. & Clark, K. E. The maternal and fetal physiologic effects of nicotine. *Semin Perinatol* **20**, 115-126, doi:10.1016/s0146-0005(96)80079-6 (1996).
- 6 Nowakowski, T. J. *et al.* Spatiotemporal gene expression trajectories reveal developmental hierarchies of the human cortex. *Science* **358**, 1318-1323, doi:10.1126/science.aap8809 (2017).
- 7 Uwada, J. *et al.* PNU-120596, a positive allosteric modulator of alpha7 nicotinic acetylcholine receptor, directly inhibits p38 MAPK. *Biochem Pharmacol* **182**, 114297, doi:10.1016/j.bcp.2020.114297 (2020).
- 8 Mayer, S. *et al.* Multimodal Single-Cell Analysis Reveals Physiological Maturation in the Developing Human Neocortex. *Neuron* **102**, 143-158 e147, doi:10.1016/j.neuron.2019.01.027 (2019).
- 9 Shen, J. X. & Yakel, J. L. Nicotinic acetylcholine receptor-mediated calcium signaling in the nervous system. *Acta Pharmacol Sin* **30**, 673-680, doi:10.1038/aps.2009.64 (2009).
- 10 Aryal, S. P. *et al.* Nicotine induces morphological and functional changes in astrocytes via nicotinic receptor activity. *Glia* **69**, 2037-2053, doi:10.1002/glia.24011 (2021).

- 11 Huang, B. *et al.* Demyelination in the medial prefrontal cortex by withdrawal from chronic nicotine causes impaired cognitive memory. *Prog Neuropsychopharmacol Biol Psychiatry* **129**, 110901, doi:10.1016/j.pnpbp.2023.110901 (2024).
- 12 Wang, L. *et al.* Molecular and cellular dynamics of the developing human neocortex. *Nature*, doi:10.1038/s41586-024-08351-7 (2025).
- 13 Zhang, J. *et al.* Isoforms of wild type proteins often appear as low molecular weight bands on SDS-PAGE. *Biotechnol J* **9**, 1044-1054, doi:10.1002/biot.201400072 (2014).
- 14 Zhang, K. *et al.* ACTB and GAPDH appear at multiple SDS-PAGE positions, thus not suitable as reference genes for determining protein loading in techniques like Western blotting. *Open Life Sci* **16**, 1278-1292, doi:10.1515/biol-2021-0130 (2021).
- 15 Guo, Y. *et al.* A diet high in sugar and fat influences neurotransmitter metabolism and then affects brain function by altering the gut microbiota. *Transl Psychiatry* **11**, 328, doi:10.1038/s41398-021-01443-2 (2021).
- 16 Qin, C. *et al.* CHRN2 represses pancreatic cancer migration and invasion via inhibiting beta-catenin pathway. *Cancer Cell Int* **22**, 340, doi:10.1186/s12935-022-02768-8 (2022).
- 17 Imamura, Y. *et al.* Ultrasound stimulation of the vagal nerve improves acute septic encephalopathy in mice. *Front Neurosci* **17**, 1211608, doi:10.3389/fnins.2023.1211608 (2023).
- 18 < <https://www.ptglab.com/products/CHRFAM7A-Antibody-21586-1-AP.htm>> (
- 19 Xiu, X., Puskar, N. L., Shanata, J. A., Lester, H. A. & Dougherty, D. A. Nicotine binding to brain receptors requires a strong cation-pi interaction. *Nature* **458**, 534-537, doi:10.1038/nature07768 (2009).
- 20 Cappello, S. *et al.* Mutations in genes encoding the cadherin receptor-ligand pair DCHS1 and FAT4 disrupt cerebral cortical development. *Nat Genet* **45**, 1300-1308, doi:10.1038/ng.2765 (2013).
- 21 Candy, J. M. *et al.* Evidence for the early prenatal development of cortical cholinergic afferents from the nucleus of Meynert in the human foetus. *Neurosci Lett* **61**, 91-95, doi:10.1016/0304-3940(85)90406-9 (1985).
- 22 Barrantes, G. E., Westwick, J. & Wonnacott, S. Nicotinic acetylcholine receptors in primary cultures of hippocampal neurons: pharmacology and Ca⁺⁺ permeability. *Biochem Soc Trans* **22**, 294S, doi:10.1042/bst022294s (1994).
- 23 Xu, Z. Q., Zhang, W. J., Su, D. F., Zhang, G. Q. & Miao, C. Y. Cellular responses and functions of alpha7 nicotinic acetylcholine receptor activation in the brain: a narrative review. *Ann Transl Med* **9**, 509, doi:10.21037/atm-21-273 (2021).
- 24 Rathouz, M. M. & Berg, D. K. Synaptic-type acetylcholine receptors raise intracellular calcium levels in neurons by two mechanisms. *J Neurosci* **14**, 6935-6945, doi:10.1523/JNEUROSCI.14-11-06935.1994 (1994).
- 25 Gueorguiev, V. D., Zeman, R. J., Meyer, E. M. & Sabban, E. L. Involvement of alpha7 nicotinic acetylcholine receptors in activation of tyrosine hydroxylase and dopamine beta-hydroxylase gene expression in PC12 cells. *J Neurochem* **75**, 1997-2005, doi:10.1046/j.1471-4159.2000.0751997.x (2000).
- 26 Ilnatovych, I. *et al.* iPSC model of CHRFAM7A effect on alpha7 nicotinic acetylcholine receptor function in the human context. *Transl Psychiatry* **9**, 59, doi:10.1038/s41398-019-0375-z (2019).
- 27 Mukhtar, T. *et al.* Temporal and sequential transcriptional dynamics define lineage shifts in corticogenesis. *EMBO J* **41**, e111132, doi:10.15252/embj.2022111132 (2022).
- 28 Paolino, A. *et al.* Differential timing of a conserved transcriptional network underlies divergent cortical projection routes across mammalian brain evolution. *Proc Natl Acad Sci U S A* **117**, 10554-10564, doi:10.1073/pnas.1922422117 (2020).
- 29 Bhaduri, A. *et al.* Cell stress in cortical organoids impairs molecular subtype specification. *Nature* **578**, 142-148, doi:10.1038/s41586-020-1962-0 (2020).
- 30 Mukhtar, T. *et al.* Tead transcription factors differentially regulate cortical development. *Sci Rep* **10**, 4625, doi:10.1038/s41598-020-61490-5 (2020).
- 31 Zhao, Y., Zhou, W., Xue, L., Zhang, W. & Zhan, Q. Nicotine activates YAP1 through nAChRs mediated signaling in esophageal squamous cell cancer (ESCC). *PLoS One* **9**, e90836, doi:10.1371/journal.pone.0090836 (2014).
- 32 Ben, Q. *et al.* A nicotine-induced positive feedback loop between HIF1A and YAP1 contributes to epithelial-to-mesenchymal transition in pancreatic ductal adenocarcinoma. *J Exp Clin Cancer Res* **39**, 181, doi:10.1186/s13046-020-01689-6 (2020).

REVIEWER COMMENTS (Black font)
REBUTTAL COMMENTS (Purple font)

Second Round:

REVIEWERS' COMMENTS

Reviewer #1 (Remarks to the Author):

The authors have responded my queries and presented new supporting evidence. The manuscript has now the quality for publication in Nature Communications.

Reviewer #2 (Remarks to the Author):

I appreciated the authors' efforts in thoroughly addressing the comments. The manuscript has been significantly improved, and I fully support its publication.

Reviewer #3 (Remarks to the Author):

The authors provide a rebuttal with most answers to my questions. This also includes an update of existing figures and text modifications. As a last minor request, the authors could make a stronger point regarding the physiology of the receptors in cortical progenitors by relying on previous study from the lab and referring to the Mayer paper in the first paragraph of the results.

We thank the reviewer for their suggestion. We have incorporated these changes in the revised manuscript.

Reviewer #4 (Remarks to the Author):
